# QUANTIZED GRADIENT PROJECTION FOR MEMORY-EFFICIENT CONTINUAL LEARNING

**Dongjun Kim**[1]**, Seohyeon Cha**[1]**, Huancheng Chen**[1]**, Chianing Wang**[2]**, Haris Vikalo**[1]
[1]University of Texas at Austin    [2]Toyota InfoTech Lab USA
{dongjungim20, seohyeon.cha, huanchengch, hvikalo}@utexas.edu,
johnny.wang@toyota.com

## ABSTRACT

Real-world deployment of machine learning models requires the ability to continually learn from non-stationary data while preserving prior knowledge and user privacy. Therefore, storing knowledge acquired from past data in a resource- and privacy-friendly manner is a crucial consideration in determining their viability. We introduce Quantized Gradient Projection Memory (QGPM), a systematic framework for continual learning that compresses and preserves the previous gradient subspace. QGPM integrates three key components: (i) distribution-aware, basis-wise quantization to minimize storage overhead, (ii) a Quantization Error-Aware (QEA) gradient projection that selectively relaxes orthogonality to mitigate gradient drift caused by accumulated quantization noise, and (iii) an on-the-fly sparse sketching strategy that improves runtime memory and computational efficiency. Experiments across multiple benchmarks demonstrate that QGPM achieves state-of-the-art performance under fixed memory budgets, highlighting its effectiveness in scalable, privacy-preserving continual learning. Our code is available here.

## 1 INTRODUCTION

Deep neural networks (DNNs) have achieved remarkable success across domains, particularly in computer vision (He et al., 2015; Simonyan & Zisserman, 2014; He et al., 2017). However, their standard training paradigm assumes access to the full dataset upfront—an unrealistic assumption in real-world scenarios where data arrives sequentially and evolves over time. In contrast, intelligent agents must learn continuously: acquiring new skills while retaining prior knowledge. This motivates the field of *continual learning* (CL) (Lange et al., 2019), which aims to enable models to adapt to new tasks without overwriting existing representations. Naively applying stochastic gradient descent (SGD) in this setting leads to *catastrophic forgetting* (McCloskey & Cohen, 1989), where learning new tasks disrupts previously acquired knowledge. To mitigate this, several CL strategies have been proposed, including: (1) Regularization-based methods (Kirkpatrick et al., 2017), which penalize updates to important weights; (2) Expansion-based methods (Rusu et al., 2016), which allocate new parameters per task; (3) Rehearsal-based methods (Rolnick et al., 2019), which replay stored or synthetic data (Gao & Liu, 2023); and (4) Projection-based methods (Saha et al., 2021; Farajtabar et al., 2020), which constrain updates to subspaces orthogonal to prior gradients. Each strategy encodes prior knowledge in a specific form: importance weights, dedicated modules, rehearsal buffers or generative models, and gradient subspaces, respectively. As Rebuffi et al. (2017) emphasize, a viable incremental learner must keep memory and compute demands bounded or slowly growing. Thus, memory efficiency is a key factor in determining a CL method's practical deployability.

Gradient Projection Memory (Saha et al., 2021) maintains a dedicated memory structure – the GPM – where it stores a set of core bases that span the gradient subspaces associated with previous tasks. When learning a new task, incoming gradients are projected onto the orthogonal complement of the subspace spanned by the stored bases, thus minimizing interference with previously acquired knowledge. These projection-based methods demonstrate state-of-the-art stability against catastrophic forgetting (Liang & Li, 2024). Furthermore, the method's inherent privacy-preserving nature, stemming from not storing raw data or intermediate representations, makes it well-suited for continual learning in privacy-sensitive fields such as the medical domain (Verma et al., 2023).

In this paper, we demonstrate that the memory efficiency of continual learning can be significantly improved by quantizing the bases stored in GPM, while preserving its core benefits. However, this introduces two key challenges: (1) the distribution of individual bases is often heavy-tailed, leading to large quantization errors; and (2) subspace distortion can cause projected gradients to deviate from their intended direction, resulting in what we call a *gradient drift*. To overcome these issues, we propose two complementary techniques: Centered Inlier Normal Float (CINF) quantization, which reduces the influence of outliers during quantization, and Quantization Error-Aware (QEA) gradient projection, which adaptively relaxes orthogonality constraints based on estimated deviation from the desired update direction.

The main contributions of this paper are:

1. We propose *Quantized Gradient Projection Memory* (QGPM), a novel framework for memory-efficient continual learning that leverages basis-wise quantization. Its core component, *Centered Inlier Normal Float* (CINF) quantization, enhances robustness to outliers and improves codebook utilization.

2. To mitigate performance degradation caused by accumulated quantization errors, we introduce *QEA gradient projection*, a technique that balances orthogonality with alignment to the desired gradient direction.

3. We propose *On-the-Fly Sparse Sketching* to accelerate the SVD computation and reduce intermediate training-time memory overhead during QGPM construction.

4. We conduct extensive experiments demonstrating that QGPM achieves strong performance under tight memory budgets, with a detailed analysis of its memory-accuracy tradeoffs.

## 2 RELATED WORK

**Continual Learning.** To address catastrophic forgetting, numerous continual learning (CL) methods have been proposed. Among these, *regularization-based methods* add constraints to prevent significant changes to parameters deemed important for earlier tasks. For example, Elastic Weight Consolidation (EWC) (Kirkpatrick et al., 2017) adds a quadratic penalty term weighted by the Fisher Information Matrix, while Synaptic Intelligence (SI) (Zenke et al., 2017) computes parameter importance during training and penalizes deviations proportionally. Despite their simplicity, such methods often suffer from low stability and require parameter importance masks that scale with model size, limiting memory efficiency. *Expansion-based methods* allocate task-specific parameters or sub-networks to isolate learning across tasks. Progressive Neural Networks (PNN) (Rusu et al., 2016) grow the model by adding new columns of neurons for each task while freezing earlier ones, whereas Hard Attention to the Task (HAT) (Serrà et al., 2018) learns task-specific binary masks to protect important neurons. While effective, these approaches incur linearly growing memory overhead, violating the goal of bounded resource usage in CL. *Rehearsal-based methods* store and replay data or intermediate features from previous tasks to reinforce memory. Experience Replay (ER) (Rolnick et al., 2019) maintains a buffer of past samples and interleaves them with current training data. DER++ (Buzzega et al., 2020) and FDR (Benjamin et al., 2019) combine rehearsal with distillation-based regularization, penalizing shifts in logits or model outputs. These methods can perform well given a moderate buffer size but raise privacy concerns due to the storage of raw data. Finally, gradient projection–based methods constrain updates to lie in subspaces orthogonal to previous gradients. OGD (Farajtabar et al., 2020) and OWM (Zeng et al., 2019) pioneered this idea, while GPM (Saha et al., 2021) improves scalability by projecting in the space of input representations. Recent works (Yang et al., 2023; Liang & Li, 2023; Lin et al., 2022; Chen et al., 2024) build on this foundation to enhance performance and efficiency.

**Memory-Efficient Continual Learning.** Memory-efficient approaches to incremental learning remain relatively underexplored. Parameter isolation via pruning has been used to allocate disjoint parameter subsets per task (Zhao et al., 2022). Iscen et al. (2020) reduces the rehearsal buffer by storing compact representations (e.g., lightweight features) instead of raw data, combined with knowledge distillation. However, storing intermediate features remains vulnerable to inversion attacks (Jacobsen et al., 2018). While Zhou et al. (2023) expands only the last few task-specific layers, the overall memory footprint still grows linearly with the number of tasks. To eliminate memory buffers altogether, data-free knowledge distillation has been proposed (Smith et al., 2021; Chung

et al., 2022; Choi et al., 2021). Although it effectively addresses privacy concerns, a full copy of the network must be retained for distillation, leading to additional memory overhead. Ermis et al. (2022) utilizes adapters on top of pretrained transformers, in conjunction with knowledge distillation; however, this is only applicable in scenarios where a pretrained foundation model is available. Many existing works leverage knowledge distillation to reduce memory usage from rehearsal or model expansion. However, this typically requires either a dedicated auxiliary network or a full copy of the backbone model, introducing significant memory and computational overhead – an aspect often overlooked when reporting overall memory budgets. To the best of our knowledge, no prior work has explored the use of quantization to compress gradient subspaces, despite its potential to offer strong privacy guarantees and improved memory efficiency.

## 3 QUANTIZED GRADIENT PROJECTION MEMORY (QGPM)

In this section, we present the core ideas behind our approach to memory-efficient continual learning via basis-wise quantization of Gradient Projection Memory (GPM). The main challenge lies in the fact that the subspace spanned by the stored bases serves as the reference for orthogonal gradient updates, making it highly sensitive to quantization-induced distortions. As shown in Theorem 3.2, the deviation between the ideal orthogonal update – computed based on the true subspace – and the update derived from a quantized subspace grows quadratically with the quantization error. We find that standard linear quantization schemes are inadequate in this setting. In response, we propose a set of techniques to mitigate quantization effects while preserving the core functionality of GPM. Algorithm 1 summarizes the full QGPM procedure, which integrates CINF quantization, QEA gradient projection, and On-the-Fly sparse sketching into a unified CL framework.

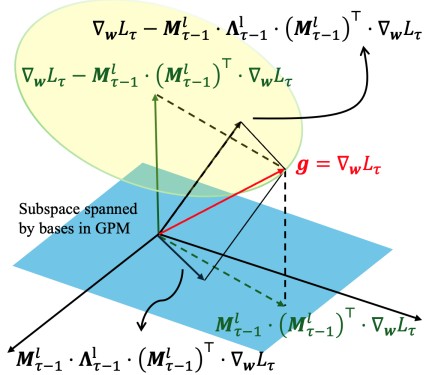

Figure 1: QEA Gradient Projection.

### 3.1 PRELIMINARIES

**Gradient Projection for Continual Learning.** The objective of continual learning is to solve $\min_{\mathbf{w}} \frac{1}{T} \sum_{\tau=1}^{T} L_\tau(\mathbf{w}, \mathcal{D}_{(\tau)})$, where $L_\tau$ and $\mathcal{D}_{(\tau)} = \{\mathcal{X}_{(\tau)}, \mathcal{Y}_{(\tau)}\}$ denote empirical loss and dataset of the $\tau$-th task. Let $\mathbf{w}_\tau^l \subset \mathbf{w}_\tau$ denote $l$-th layer parameters trained on task $\tau$ and let $\mathbf{R}_\tau^l$ be the input representation to layer $l$ at that point. The output activation of layer $l$ after training on task $\tau$ is given by $\mathbf{w}_\tau^l \cdot \mathbf{R}_\tau^l$. We would like this activation to remain unchanged even after learning task $\tau + 1$, i.e., $\mathbf{w}_\tau^l \cdot \mathbf{R}_\tau^l = \mathbf{w}_{\tau+1}^l \cdot \mathbf{R}_\tau^l = (\mathbf{w}_\tau^l + \Delta \mathbf{w}^l) \cdot \mathbf{R}_\tau^l$. This requires that the weight update $\Delta \mathbf{w}^l$ satisfies the orthogonality condition $\Delta \mathbf{w}^l \cdot \mathbf{R}_\tau^l = 0$. To enforce this constraint, we store a set of core basis vectors that span the subspace of $\mathbf{R}_\tau^l$ (see lines 13–16 of Algorithm 3 in Appendix B), and restrict future gradient updates to lie in the orthogonal complement of this subspace (see line 5 of the same algorithm). We denote this memory (i.e., GPM) for layer $l$ after task $\tau$ by $\mathbf{M}_\tau^l$.

**$k$-bit NormalFloat (NFk) Quantization.** The authors of QLoRA (Dettmers et al., 2023) introduced NFk, an information-theoretically optimal quantization data type for normally distributed data. NFk is a variant of quantile quantization, which ensures that each quantization bin contains an equal number of data points mapped from the original distribution. For $k$-bit quantization, NFk stores $2^k$ codes in a pre-defined codebook: $\mathbf{c} = [c_1, c_2, \ldots, c_{2^k}] \in [-1, 1]^{2^k}$. Given a vector $\mathbf{u} = (a_1, a_2, \ldots, a_B) \in \mathbb{R}^B$, the quantization proceeds as follows: (1) Compute $M = \max_i |a_i|$. (2) Normalize each entry by $M$ to scale the values into the range $[-1, 1]$. (3) For each normalized value, find the index $j$ of the closest codebook value $c_j$, i.e., $a_i$ is mapped/discretized according to the index $j = \mathrm{argmin}_j |c_j - \frac{a_i}{M}|$.

### 3.2 CENTERED INLIER NORMAL FLOAT (CINF)

A key limitation of naïve NFk quantization is its vulnerability to outliers. It normalizes the input vector by its maximum absolute value, which can become large in the presence of outliers. This

compresses the bulk of values toward zero, distorting the distribution into a highly sub-Gaussian. As a result, the quantization codebook is used inefficiently: most values are mapped to bins near zero, while only a few occupy the extreme bins near $-1$ and $1$, leading to significant information loss (Yoshida, 2023). To address this issue, we propose Centered Inlier Normal Float (CINF) quantization, which improves robustness to outliers by restricting normalization to a central quantile range. Given an input vector $\mathbf{u} = (a_1, a_2, \ldots, a_B)$, CINF performs the following steps:

1. Compute the mean of the input, $\mu = \frac{1}{B} \sum_{i=1}^{B} a_i$.
2. Center the vector components by subtracting the mean, $a_{i,\text{centered}} = a_i - \mu$.
3. Compute the $\delta$ and $1 - \delta$ quantiles of the centered values, $q_\delta$ and $q_{1-\delta}$.
4. Values outside the quantile range $[q_\delta, q_{1-\delta}]$ are stored in full precision. The remaining inlier values are quantized using NFk, where normalization is performed using the scale factor $s = \max(|q_\delta|, |q_{1-\delta}|)$ rather than the global maximum $M = \max_i |a_i|$.

Further details on CINF codebook construction are provided in Appendix E.1. We now analyze how this method reduces quantization error. Assume entries of vector $\mathbf{u}$ follow a standard normal distribution, i.e., $a_i \sim \mathcal{N}(0, 1)$. Let $M = \max_i |a_i|$, and define the quantile threshold $q_\delta = \inf\{r \geq 0 : \frac{1}{B} \sum_{i=1}^{B} \mathbf{1}(|a_i| \leq r) \geq \delta\}$, with scaling factor $s = \max(|q_\delta|, |q_{1-\delta}|)$. Partition the data into outliers $\mathcal{O} = \{i : |a_i| \geq s\}$ and inliers $\mathcal{I} = \mathcal{O}^C$. Let $\mathbf{c} = [c_1, c_2, \ldots, c_{2^k}]$ denote the fixed NFk codebook. The dequantized value under naïve NFk is

$$\tilde{a}_i^{\text{NF}k} = M \, \mathbf{c}_{j^*(i)}, \qquad j^*(i) = \underset{j \in \{1, \ldots, 2^k\}}{\arg\min} \left| \mathbf{c}_j - \frac{a_i}{M} \right|, \qquad i \in \mathcal{O} \cup \mathcal{I}. \qquad (1)$$

In contrast, the dequantized value under CINF is given by

$$\tilde{a}_i^{\text{CINF}} = \begin{cases} a_i, & i \in \mathcal{O}, \\ s \, \mathbf{c}_{j^*(i)} + \mu, & i \in \mathcal{I}, \end{cases} \qquad j^*(i) = \underset{j \in \{1, \ldots, 2^k\}}{\arg\min} \left| \mathbf{c}_j - \frac{a_i}{s} \right|. \qquad (2)$$

The quantization errors of NFk and CINF can be respectively modeled as:

$$e^{\text{NF}k} = \frac{1}{|\mathcal{O} \cup \mathcal{I}|} \sum_{i \in (\mathcal{O} \cup \mathcal{I})} (a_i - \tilde{a}_i^{\text{NF}k})^2, \quad e^{\text{CINF}} = \frac{1}{|\mathcal{I}|} \sum_{i \in \mathcal{I}} (a_i - \tilde{a}_i^{\text{CINF}})^2, \qquad (3)$$

where the CINF error ($e^{\text{CINF}}$) is computed over inliers only, as outliers are stored losslessly. To understand how normalization impacts quantization error, we compare the NFk scale factor $M = \max_i |a_i|$ to the CINF quantile threshold $q_\delta$. The next result from order statistics offers a useful approximation for this comparison.

**Theorem 3.1 (Expected Values of Normal Order Statistics (Harter, 1961))** *Consider* $\mathbf{u} = (a_1, a_2, \ldots, a_B)$, *a set of independent random variables drawn from a standard normal distribution, i.e., $a_i \sim \mathcal{N}(0, 1)$. Denote the expected value of the $i$-th order statistic by $\mathbb{E}[i : B]$. Then, for sufficiently large $B$, this expectation can be approximated as:*

$$\mathbb{E}[i : B] \approx \Phi^{-1}\left(\frac{i - \alpha}{B - 2\alpha + 1}\right), \qquad (4)$$

*where $\Phi^{-1}$ is the inverse cumulative distribution function (CDF) of $\mathcal{N}(0, 1)$, and $\alpha = 0.375$.*

By definition, $\mathbb{E}[M] = \mathbb{E}[B : B] \approx \Phi^{-1}(\frac{B - \alpha}{B - 2\alpha + 1})$. As $B$ grows, $M$ becomes large since $\frac{B - \alpha}{B - 2\alpha + 1} \to 1$. As previously discussed, this leads to overly aggressive normalization of the data via $a_i/M$, compressing most values near zero and resulting in inefficient utilization of quantization bins. In contrast, by excluding a small fraction of outliers (e.g., the top 1%), the expected scale for inlier-based normalization becomes $\mathbb{E}[s] = \mathbb{E}[0.99B : B] \approx \Phi^{-1}(\frac{0.99B - \alpha}{B - 2\alpha + 1})$ which remains bounded above by $\Phi^{-1}(0.99) = 2.32$. This results in a more evenly spread normalized distribution and improved codebook utilization compared to the NFk case. Consequently, CINF produces a lower quantization error than NFk:

$$e^{\text{CINF}} \leq e^{\text{NF}k}. \qquad (5)$$

---

**Algorithm 1** QGPM Training Algorithm

---

**Input:** $f_{\mathbf{w}}$ the NN model, $\mathcal{D}^{train}$ the training dataset, $\eta$ the learning rate, and $\epsilon_{th}$ the threshold value. Initialize, $\mathcal{M}_{Q,0}^l, \mathcal{S}_0^l, \mathcal{Z}_0^l, \mathcal{O}_0^l \leftarrow [\,]$, for all $l = 1, 2, \ldots, L'$, and $\mathbf{w} \leftarrow \mathbf{w}_o$.

1: **for** $\tau = 1, 2, \ldots, T$ **do**
2:    **for** $\forall l \in \{1, 2, \ldots, L'\}$ **do**
3:       $\mathbf{M}_{\tau-1}^l \leftarrow \mathrm{Dequant}(\mathcal{M}_{Q,\tau-1}, \mathcal{S}_{\tau-1}^l, \mathcal{Z}_{\tau-1}^l)$
4:       $\mathbf{P}_{\tau-1}^l \leftarrow \mathbf{M}_{\tau-1}^l \cdot \mathrm{Diag}(\mathcal{O}_{\tau-1}^l) \cdot (\mathbf{M}_{\tau-1}^l)^\top$
5:    **end for**
6:    **repeat**
7:       $B_n \sim \mathcal{D}_\tau^{train}$
8:       $\mathbf{g} \leftarrow \nabla_{\mathbf{w}} L_\tau$
9:       $\hat{\mathbf{g}} \leftarrow \mathbf{g} - \mathbf{P}_{\tau-1} \cdot \mathbf{g}$
10:      $\mathbf{w} \leftarrow \mathbf{w} - \eta \cdot \hat{\mathbf{g}}$
11:    **until** convergence
12:    $B_{n_s} \sim \mathcal{D}_\tau^{train}$
13:    $\{\mathbf{R}_\tau^l\}_{l=1}^{L'} \leftarrow \mathrm{forward}(B_{n_s}, f_{\mathbf{w}})$
14:
15:    **for** $\forall l \in \{1, 2, \ldots, L'\}$ **do**
16:       $\hat{\mathbf{R}}_\tau^l \leftarrow \mathbf{R}_\tau^l - \mathbf{M}_{\tau-1}^l \cdot (\mathbf{M}_{\tau-1}^l)^\top \cdot \mathbf{R}_\tau^l$
17:       $\hat{\mathbf{U}}_\tau^l, \hat{\boldsymbol{\Sigma}}_\tau^l, \hat{\mathbf{V}}_\tau^l \leftarrow \mathrm{SVD}(\hat{\mathbf{R}}_\tau^l)$
18:       $r \leftarrow \mathrm{criterion}(\hat{\mathbf{R}}_\tau^l, \mathbf{R}_\tau^l, \epsilon_{th})$
19:       $\mathbf{U}_{Q,\tau}^l, \mathbf{s}_\tau^l, \mathbf{z}_\tau^l, \mathbf{o}_\tau^l \leftarrow \mathrm{Quant}(\hat{\mathbf{U}}_\tau^l[\cdot, 1:r])$
20:       $\mathcal{M}_{Q,\tau} \leftarrow \mathrm{Concat}[\mathcal{M}_{Q,\tau-1}, \mathbf{U}_{Q,\tau}^l]$
21:       $\mathcal{S}_\tau^l \leftarrow \mathrm{Concat}[\mathcal{S}_{\tau-1}^l, \mathbf{s}_\tau^l]$
22:       $\mathcal{Z}_\tau^l \leftarrow \mathrm{Concat}[\mathcal{Z}_{\tau-1}^l, \mathbf{z}_\tau^l]$
23:       $\mathcal{O}_\tau^l \leftarrow \mathrm{Concat}[\mathcal{O}_{\tau-1}^l, \mathbf{o}_\tau^l]$
24:    **end for**
25: **end for**

---

Now, we define the basis-wise CINF quantization and dequantization functions for a single input vector $\mathbf{u} \in \mathbb{R}^d$ as $(\mathbf{u}_q, s, \mu, \lambda) = \mathcal{Q}_{\mathrm{CINF}}(\mathbf{u})$ and $\tilde{\mathbf{u}} = \mathcal{Q}_{\mathrm{CINF}}^{-1}(\mathbf{u}_q, s, \mu)$, where $\mathbf{u}_q$ is the quantized vector, $s$ is the inlier scale, $\mu$ is the mean, and $\lambda$ is orthogonality weight (introduced later). In our implementation, the Quant function in Algorithm 1 applies the column-wise operator $\mathcal{Q}_{\mathrm{CINF}}$ to the top-$r$ left singular vectors $\hat{\mathbf{U}}_\tau^l[\cdot, 1:r]$ obtained from SVD. Formally,

$$\mathbf{U}_{Q,\tau}^l[\cdot, \, i], \ \mathbf{s}_\tau^l[i], \ \mathbf{z}_\tau^l[i], \ \mathbf{o}_\tau^l[i] = \mathcal{Q}_{\mathrm{CINF}}(\hat{\mathbf{U}}_\tau^l[\cdot, \, i]), \quad \forall i \in \{1, \ldots, r\}. \tag{6}$$

Here, $\mathbf{U}_{Q,\tau}^l = [\mathbf{u}_q^{(1)}, \ldots, \mathbf{u}_q^{(r)}] \in \mathbb{R}^{d \times r}$ stores the quantized basis vectors, while the auxiliary arrays $\mathbf{s}_\tau^l = [s_1, \ldots, s_r]$, $\mathbf{z}_\tau^l = [\mu_1, \ldots, \mu_r]$, and $\mathbf{o}_\tau^l = [\lambda_1, \ldots, \lambda_r]$ record the corresponding scales, offsets, and orthogonality weights used in dequantization and QEA projection.

### 3.3 Quantization Error-Aware (QEA) Gradient Projection

**QEA Gradient Projection.** In standard GPM-based CL, when training on the $\tau$-th task using dataset $\mathcal{D}_{(\tau)}$, the raw gradient $\mathbf{g} = \nabla_{\mathbf{w}} L_\tau$ is computed via stochastic gradient descent (SGD). This gradient is then projected to be orthogonal to the subspace spanned by the basis vectors stored in $\mathbf{M}_{\tau-1}^l$ (see line 5, Algorithm 3 in Appendix B). In QGPM, the basis vectors are quantized to reduce memory footprint, forming $\mathcal{M}_{Q,\tau}^l$. To perform gradient projection (line 9, Algorithm 1), the quantized basis vectors must be dequantized into full-precision (or optionally, a compact higher-precision format such as BF16) for use in tensor operations. This is handled by the Dequant function in line 3 of Algorithm 1, which reconstructs each basis vector by applying column-wise $\mathcal{Q}_{\mathrm{CINF}}^{-1}$ operator

$$\mathbf{M}_{\tau-1}^l[\cdot, \, i] = \mathcal{Q}_{\mathrm{CINF}}^{-1}(\mathcal{M}_{Q,\tau-1}^l[\cdot, \, i], \, \mathcal{S}_{\tau-1}^l[i], \, \mathcal{Z}_{\tau-1}^l[i]), \quad \forall i \in \{1, \ldots, |\mathcal{M}_{Q,\tau-1}^l|\} \tag{7}$$

where $|\mathcal{M}_{Q,\tau-1}^l|$ denotes the number of basis vectors in the QGPM. However, as quantized bases are accumulated over multiple tasks, the quantization error compounds. We observe that this accumulated distortion can destabilize the learning process: when the subspace spanned by the quantized GPM is significantly misaligned from its original counterpart, the resulting gradient projection deviates from the true orthogonal direction. This leads to update directions that are no longer interference-free with respect to past tasks – we refer to these as gradient drift.

**Theorem 3.2 (Quantization Error Accumulation)** *Let* $\mathbf{M}_o = [\mathbf{u}_1, \ldots, \mathbf{u}_m] \in \mathbb{R}^{n \times m}$ *be the full-precision GPM, and let* $\mathbf{E} = [\boldsymbol{\epsilon}_1, \ldots, \boldsymbol{\epsilon}_m] \in \mathbb{R}^{n \times m}$ *be a random Gaussian error matrix, where each column* $\boldsymbol{\epsilon}_j \sim \mathcal{N}(\mathbf{0}, \sigma^2 \mathbf{I}_n)$. *Define the quantized GPM as* $\mathbf{M}_e = \mathbf{M}_o + \mathbf{E}$. *Let* $\mathbf{g} \in \mathbb{R}^n$ *be a gradient vector, and let* $\hat{\mathbf{g}}_o$ *and* $\hat{\mathbf{g}}_e$ *denote its orthogonal components with respect to the subspaces spanned by* $\mathbf{M}_o$ *and* $\mathbf{M}_e$, *respectively. The expected deviation between the projected gradients satisfies:*

$$\mathbb{E}\left[\|\hat{\mathbf{g}}_o - \hat{\mathbf{g}}_e\|_2\right] \geq \|\mathbb{E}\left[\hat{\mathbf{g}}_o - \hat{\mathbf{g}}_e\right]\|_2 = m \cdot \sigma^2 \cdot \|\mathbf{g}\|_2, \tag{8}$$

*implying that the projection error introduced by quantization grows linearly with the number of basis vectors $m$ and quadratically with the quantization noise level $\sigma$. The proof is provided in Appendix G.1.*

QGPM mitigates quantization errors by relaxing the strict orthogonality constraint of standard GPM, allowing a controlled parallel gradient component. This relaxation is adaptively scaled based on the quantization fidelity of each basis vector, which may vary across vectors due to differences in their distributions and compressibility. To capture this fidelity, the `Quant` function evaluates the quantization error individually for each of the $r$ new basis vectors $\hat{\mathbf{U}}^l_\tau[:, i]$ to be incorporated into the QGPM. Specifically, immediately after quantization via Eq 6, each quantized basis vector $\mathbf{U}^l_{Q,\tau}[\cdot, i]$ is dequantized to obtain an approximate reconstruction that incorporates quantization error:

$$\tilde{\mathbf{U}}^l_\tau[\cdot, i] = \mathcal{Q}^{-1}_{\text{CINF}}(\mathcal{Q}_{\text{CINF}}(\hat{\mathbf{U}}^l_\tau[\cdot, i])), \quad \forall i \in \{1, \dots, r\}. \tag{9}$$

To assess the distortion introduced by quantization, we compute the cosine similarity between each original full-precision basis vector ($\hat{\mathbf{U}}^l_\tau[:, i]$) and its dequantized counterpart ($\tilde{\mathbf{U}}^l_\tau[\cdot, i]$). Finally, the quantization error for the $i$-th basis is defined as:

$$e_i = 1 - \text{sim}_{\cos}(\tilde{\mathbf{U}}^l_\tau[\cdot, i], \hat{\mathbf{U}}^l_\tau[\cdot, i]) = 1 - \frac{\tilde{\mathbf{U}}^l_\tau[\cdot, i] \cdot \hat{\mathbf{U}}^l_\tau[\cdot, i]}{\|\tilde{\mathbf{U}}^l_\tau[\cdot, i]\|_2 \, \|\hat{\mathbf{U}}^l_\tau[\cdot, i]\|_2}. \tag{10}$$

The value $e_i$ quantifies how much the dequantized vector $\tilde{\mathbf{U}}^l_\tau[\cdot, i]$ deviated from the original basis $\hat{\mathbf{U}}^l_\tau[\cdot, i]$. To incorporate this into the gradient projection mechanism, we introduce a hyperparameter $\alpha$ that scales the error, and define an orthogonality weight $\lambda_i = 1 - \alpha \cdot e_i$. The `Quant` function computes $\lambda_i$ for each of the $r$ new basis vectors, resulting in a distinct orthogonality weight for every column in the QGPM. These weights are then used to specify the projection strength during gradient updates, controlling the extent to which the orthogonality is enforced. Formally,

$$\nabla_{\mathbf{w}} \hat{L}_\tau = \nabla_{\mathbf{w}} L_\tau - \mathbf{M}^l_{\tau-1} \cdot \Lambda^l_{\tau-1} \cdot (\mathbf{M}^l_{\tau-1})^\top \cdot \nabla_{\mathbf{w}} L_\tau = \nabla_{\mathbf{w}} L_\tau - \mathbf{P}^l_{\tau-1} \cdot \nabla_{\mathbf{w}} L_\tau, \tag{11}$$

where $\boldsymbol{\Lambda}^l_{\tau-1} = \text{Diag}(\lambda_1, \dots, \lambda_{|\mathcal{M}^l_{Q,\tau-1}|}) \in \mathbb{R}^{|\mathcal{M}^l_{Q,\tau-1}| \times |\mathcal{M}^l_{Q,\tau-1}|}$, as illustrated in Figure 1. When the quantization error $e_i$ is large, the corresponding $\lambda_i$ is reduced, allowing a greater component of the gradient to pass through in the parallel direction – thereby increasing plasticity along the associated basis vector. This adaptive mechanism mitigates the impact of projection distortion due to quantization error, effectively preserving both model stability and past knowledge.

### 3.4 QGPM CONSTRUCTION WITH ON-THE-FLY SPARSE SKETCHING

After completing the $\tau$-th task, QGPM is updated to $\mathcal{M}^l_\tau$ by incorporating $r$ new basis vectors. To do so, a subset of training samples $B_{n_s} \subset \mathcal{D}^{train}_\tau$ is passed through the network $f_{\mathbf{w}}$, producing input representations at each layer, $\mathcal{R}_\tau = [\mathbf{R}^1_\tau, \mathbf{R}^2_\tau, \dots, \mathbf{R}^L_\tau]$. We extract the orthogonal component of the new representation relative to the current GPM subspace $\mathbf{M}^l_{\tau-1}$, i.e., $\hat{\mathbf{R}}^l_\tau = \mathbf{R}^l_\tau - \mathbf{M}^l_{\tau-1} \cdot (\mathbf{M}^l_{\tau-1})^\top \cdot \mathbf{R}^l_\tau$, where $\mathbf{M}^l_{\tau-1} \cdot (\mathbf{M}^l_{\tau-1})^\top \cdot \mathbf{R}^l_\tau$ is the component that already exists in $\mathbf{M}^l_{\tau-1}$. To extract orthogonal basis vectors from $\hat{\mathbf{R}}^l_\tau$, we apply Singular Value Decomposition (SVD) and retain the top-$r$ singular vectors, $\hat{\mathbf{U}}^l_{\tau,r} \hat{\mathbf{\Sigma}}^l_{\tau,r} \hat{\mathbf{V}}^l_{\tau,r} = \hat{\mathbf{R}}^l_{\tau,r} \approx \hat{\mathbf{R}}^l_\tau$, where the value of $r$ is chosen by the function criterion:

$$\frac{\|\mathbf{M}^l_{\tau-1} \cdot (\mathbf{M}^l_{\tau-1})^\top \cdot \mathbf{R}^l_\tau\|^2_F}{\|\mathbf{R}^l_\tau\|^2_F} + \frac{\|\hat{\mathbf{R}}^l_{\tau,r}\|^2_F}{\|\mathbf{R}^l_\tau\|^2_F} \geq \epsilon_{th}. \tag{12}$$

This ensures that the retained directions collectively explain at least $\epsilon_{th}$ of the total variance in $\mathbf{R}^l_\tau$. The selected $r$ left singular vectors are then quantized, yielding $r$ quantized singular vectors $\mathbf{U}^l_{Q,\tau}$, along with auxiliary arrays, $(\mathbf{s}^l_\tau, \mathbf{z}^l_\tau, \mathbf{o}^l_\tau)$, as introduced in Section 3.2. Finally, the newly quantized basis vectors and their associated metadata are appended to the QGPM: $\mathcal{M}^l_{Q,\tau}$, $\mathcal{S}^l_\tau$, $\mathcal{Z}^l_\tau$, and $\mathcal{O}^l_\tau$.

**On-the-Fly Sparse Sketching.** In the standard GPM update, the representation matrix $\mathbf{R}^l_\tau$ is constructed by collecting local activation patches from the entire feature representation. Each patch is flattened into a column vector $\mathbf{r}_i$, and these vectors are concatenated to form $\mathbf{R}^l_\tau = [\mathbf{r}_1, \dots, \mathbf{r}_N]$. A

key challenge is that $N$ can become extremely large, especially in convolutional or transformer blocks – popular components of modern neural networks. As shown in Table 10 in Appendix C, this results in a high-dimensional matrix that significantly slows down SVD and incurs substantial memory overhead during training. To address this, we propose an On-the-Fly Sparse Sketching strategy, which constructs a low-dimensional approximation of $\mathbf{R}_\tau^l$ using a sparse sketch in a streaming manner.

**Theorem 3.3** (($1 \pm \epsilon$)-$\ell_2$ **Subspace Embedding via Sparse Sketching**) *Let* $\mathbf{S} \in \mathbb{R}^{r \times N}$ *be a sparse embedding matrix constructed using hash functions* $h : [N] \to [r]$ *and* $\sigma : [N] \to \{-1, 1\}$. *The $i$-th column of* $\mathbf{S}$ *has a single non-zero entry at row* $h(i)$, *with value* $\sigma(i)$. *Then, for any rank-$k$ matrix* $\mathbf{A} \in \mathbb{R}^{N \times n}$, *if* $r = O(\frac{k^2}{\epsilon^2 \delta})$, *then with probability at least* $1 - \delta$, $\mathbf{S}$ *is a* ($1 \pm \epsilon$)-$\ell_2$ *subspace embedding for the row space of* $\mathbf{A}$ *in the* $\ell_2$*-norm. Equivalently, for all* $x \in \mathbb{R}^n$,

$$(1 - \varepsilon)\,\|Ax\|_2 \ \leq \ \|SAx\|_2 \ \leq \ (1 + \varepsilon)\,\|Ax\|_2. \tag{13}$$

Proof is in Appendix G.2. Leveraging the ($1 \pm \epsilon$)-$\ell_2$ subspace embedding property of sparse sketching (Theorem 3.3), we apply column-wise sketching for each new representation vector $\mathbf{r}_i$ by setting $r = n$. This reduces the size of the representation matrix from $\mathbb{R}^{n \times N}$ to $\mathbb{R}^{n \times n}$, while approximately preserving the geometric structure of the original subspace. As a result, the peak intermediate memory required to construct the QGPM is reduced by a factor of $N/n$ at each layer. In addition, the computational cost of SVD, the primary bottleneck in standard GPM, is reduced by the same factor. Specifically, the time complexity of computing the SVD of an $n \times N$ matrix is $O(N \times n^2)$ when $N \gg n$. Notably, with this sketching scheme, the representation matrix for ViT-S can be compressed from 1.62GB to just 66MB. Detailed implementation and analysis are provided in Appendix C.

## 4 EXPERIMENTS

**Setup.** We evaluate QGPM on three standard CL benchmarks: 10-split CIFAR-100, 5-Datasets (Ebrahimi et al., 2020), and 10/20-split miniImageNet (Vinyals et al., 2016). These benchmarks capture complementary challenges of continual learning, including large label spaces, cross-domain heterogeneity, and scalability. We follow standard architectures for each benchmark: AlexNet (Serra et al., 2018) for CIFAR-100, ResNet-18 (He et al., 2016) for 5-Datasets, and pretrained ViT-S (Touvron et al., 2022) for miniImageNet. All models are trained in a multi-head setting, with a dedicated classification head per task. Further architectural and dataset details are in Appendix A.2.

**Baselines.** We compare QGPM against rehearsal-based continual learning methods that support flexible memory buffer sizes, unlike regularization- or expansion-based approaches. Specifically, we include Average GEM (AGEM) (Chaudhry et al., 2019), Function Distance Regularization (FDR) (Benjamin et al., 2019), Experience Replay (ER) (Rolnick et al., 2019), and Dark Experience Replay (DER++) (Buzzega et al., 2020), which represent widely used and competitive baselines. We also evaluate two GPM variants: the original full-precision version (GPM-FP) and a memory-constrained variant with reduced $\epsilon_{th}$ (GPM-FC-MC). For fairness, all methods are matched to QGPM's memory budget by adjusting the buffer size to yield an equivalent overall overhead.

**Metrics.** We evaluate performance using two standard continual learning metrics (Lopez-Paz & Ranzato, 2017): **average accuracy (ACC)** and **backward transfer (BWT)**. ACC measures the overall test accuracy across all tasks at the end of training, while the BWT quantifies the degree of forgetting (with negative values indicating performance loss on earlier tasks). They are defined as:

$$\text{ACC} = \frac{1}{T} \sum_{t=1}^{T} acc_t^T, \quad \text{BWT} = \frac{1}{T-1} \sum_{t=1}^{T-1} (acc_t^T - acc_t^t), \tag{14}$$

where $acc_t^T$ is the final accuracy on task $t$ after learning all $T$ tasks, and $acc_t^t$ is the accuracy on task $t$ immediately after it is learned.

### 4.1 PERFORMANCE ANALYSIS

**Main Results.** Table 1 reports results for 8-bit QGPM (8QGPM) and 4-bit QGPM (4QGPM) compared to all baselines. Across benchmarks, 8QGPM matches full-precision GPM (GPM-FP), with ACC and BWT reduced by less than $0.5\%$, while requiring $3.83\times$ less memory on average. Under

Table 1: Performance comparison on continual learning benchmarks. 8-bit QGPM (8QGPM) matches full-precision GPM (GPM-FP) with <0.5% drop in ACC/BWT while reducing memory by $3.83\times$ on average. Under the same memory budget, rehearsal-based baselines (AGEM, FDR, ER, DER++) suffer substantial forgetting (higher BWT). 4-bit QGPM (4QGPM) remains robust (<1% drop vs. GPM-FP) with a $6.44\times$ memory reduction, demonstrating scalability under stricter constraints.

| Methods | 10-split CIFAR100 | | 5-Datasets | | 10-split miniImageNet | | 20-split miniImageNet | |
|---|---|---|---|---|---|---|---|---|
| | ACC ($\uparrow$) | BWT ($\uparrow$) | ACC ($\uparrow$) | BWT ($\uparrow$) | ACC ($\uparrow$) | BWT ($\uparrow$) | ACC ($\uparrow$) | BWT ($\uparrow$) |
| Memory | 3.13 MB | | 27.67 MB | | 9.85 MB | | 11.07 MB | |
| GPM-FP | 71.11 | -0.98 | 89.52 | -1.83 | 73.84 | -3.30 | 80.28 | -2.73 |
| Memory | 0.81 MB | | 7.24 MB | | 2.59 MB | | 2.88 MB | |
| GPM-FP-MC | 64.58 | -12.88 | 77.82 | -16.14 | 71.48 | -8.24 | 77.06 | -6.69 |
| AGEM-MC | 50.72 | -25.44 | 80.45 | -14.81 | 57.43 | -23.74 | 60.42 | -24.02 |
| DER++-MC | 60.63 | -14.82 | 84.08 | -8.10 | 68.77 | -11.51 | 76.12 | -9.35 |
| ER-MC | 56.41 | -16.77 | 80.97 | -11.79 | 67.95 | -14.37 | 75.56 | -10.08 |
| FDR-MC | 63.07 | -13.74 | 83.54 | -9.88 | 67.80 | -12.93 | 70.73 | -13.94 |
| **8QGPM (Ours)** | **70.70** | **-0.81** | **89.41** | **-2.08** | **73.78** | **-3.54** | **80.23** | **-2.77** |
| Memory | 0.48 MB | | 4.27 MB | | 1.52 MB | | 1.76 MB | |
| GPM-FP-MC | 62.33 | -20.23 | 70.86 | -25.15 | 70.90 | -9.48 | 76.36 | -7.64 |
| **4QGPM (Ours)** | **69.74** | **-2.62** | **88.51** | **-3.79** | **73.77** | **-4.50** | **80.25** | **-3.38** |

the same memory budget, rehearsal-based baselines suffer substantial forgetting due to limited buffer size, leading to much higher BWT. With more aggressive compression, 4QGPM shows only a modest drop ($< 1\%$ ACC/BWT relative to GPM-FP) while achieving a $6.44\times$ memory reduction. Since 8QGPM already outperforms all baselines at equal memory, we focus on the comparisons involving 8-bit results; 4-bit results highlight the scalability of QGPM under stricter memory constraints.

**Memory Profile.** Figure 2(a) reports the memory footprint of QGPM compared to full-precision GPM (GPM-FP), with model parameter sizes included for reference. Specifically, the maximum memory footprint of the fully occupied GPM can be compressed as follows: (1) ViT-S: 66.3MB $\rightarrow$ 17.8MB (8-bit) / 10.2MB (4-bit); (2) ResNet-18: 34.27MB $\rightarrow$ 8.93MB (8-bit) / 5.29MB (4-bit); and (3) AlexNet: 22.27MB $\rightarrow$ 5.74MB (8-bit) / 3.41MB (4-bit). These reductions approach the theoretical $4\times$ (8-bit) and $8\times$ (4-bit) limits, with minor deviation due to overhead from storing auxiliary metadata ($\mathcal{S}_\tau^l, \mathcal{Z}_\tau^l, \mathcal{O}_\tau^l$) and full-precision outliers. Additional experimental setup and hyperparameter details can be found in Appendix A.

**Runtime Analysis.** Figure 2(b) reports normalized training time under matched memory budgets for AlexNet, ResNet-18, and ViT-S. Sparse sketching is applied to GPM-FP_s and 8QGPM_s, but not GPM-FP. SGD is included as a lower bound, since it does not incorporate any continual learning mechanisms. On AlexNet and ResNet-18, GPM-family models achieve the lowest training time, as their projection operation is cheaper than the additional forward/backward passes and distillation losses required in rehearsal-based baselines. The quantization/dequantization overhead in QGPM is negligible, as shown by the similar runtimes of 8QGPM_s and GPM-FP_s. For ViT-S, however, GPM-FP exhibits a significant slowdown because the SVD on large representations of transformer blocks dominate training cost. Applying sparse sketching mitigates this bottleneck, accelerating both SVD and overall training.

Table 2: Effect of quantization bitwidth on QGPM.

| bit | $e_{avg}$ | $e_{max}$ | ACC(%) | BWT |
|---|---|---|---|---|
| 4-bit | 0.544 | 1.173 | 25.03 | -31.27 |
| **5-bit** | **0.126** | **0.254** | **38.97** | **-22.94** |
| 6-bit | 0.033 | 0.057 | 64.22 | -0.32 |
| 8-bit | 0.004 | 0.011 | 65.01 | 0.61 |
| FP | 0 | 0 | 65.02 | 0.59 |

Table 3: Effect of outlier proportion on QGPM.

| $p(\%)$ | $e_{avg}$ | $e_{max}$ | ACC(%) | BWT |
|---|---|---|---|---|
| 0.0 | 0.573 | 1.205 | 41.23 | -24.40 |
| **0.5** | **0.434** | **0.650** | **49.71** | **-16.29** |
| 1.0 | 0.401 | 0.553 | 50.26 | -15.84 |
| 1.5 | 0.373 | 0.485 | 55.02 | -11.76 |
| 2.0 | 0.353 | 0.453 | 55.36 | -11.07 |

Table 4: Effect of QEA scaling factor $\alpha$ on $\lambda_{avg}$, ACC, and BWT. Moderate $\alpha$ improves performance by relaxing orthogonality based on quantization error.

| $\alpha$ | $\lambda_{avg}$ | ACC(%) | BWT |
|---|---|---|---|
| SGD | - | 59.77 | -18.68 |
| FP | - | 65.13 | -1.07 |
| 0 | 1.0 | 44.20 | -20.69 |
| 10 | 0.962 | 40.54 | -25.44 |
| 40 | 0.849 | 58.73 | -8.96 |
| 60 | 0.775 | 65.58 | -3.87 |
| **80** | **0.699** | **67.26** | **-2.91** |
| 100 | 0.622 | 66.59 | -4.24 |
| 150 | 0.433 | 65.91 | -5.79 |
| 200 | 0.249 | 65.17 | -7.28 |

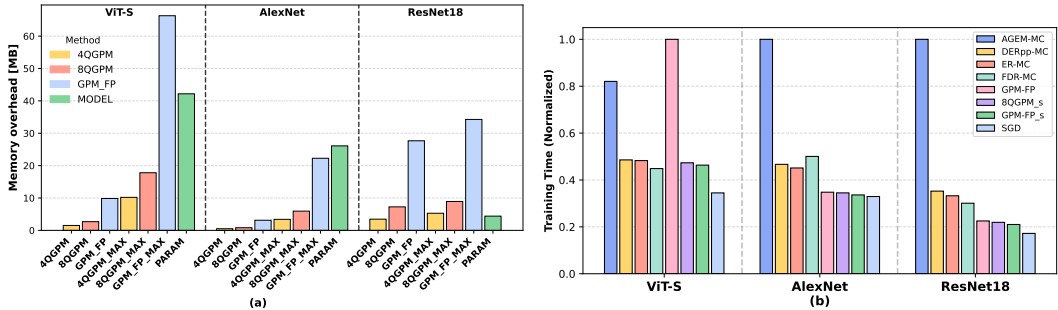

Figure 2: (a) Memory profile of QGPM vs. full-precision GPM. QGPM approaches the 4× (8-bit) and 8× (4-bit) compression limits with only minor metadata overhead, yielding substantial savings across all architectures. (b) Normalized training time under matched memory budgets. QGPM adds negligible overhead compared to GPM-FP, and sparse sketching mitigates the ViT-S bottleneck by accelerating SVD operations.

## 4.2 EFFECT OF BITWIDTH AND OUTLIERS ON QUANTIZATION FIDELITY

We examine how quantization bitwidth and the outlier proportion influence error characteristics and, ultimately, QGPM performance by analyzing experiments on 10-split CIFAR-100. To this end, we introduce two metrics to quantify quantization error: **average quantization error** across all stored bases ($e_{avg}$) and **average per-layer maximum error** ($e_{max}$), defined respectively as:

$$e_{avg} = \frac{1}{\sum_{l=1}^{L} |\mathcal{M}_{Q,T}^l|} \sum_{l=1}^{L} \sum_{i=1}^{|\mathcal{M}_{Q,T}^l|} e_i^l, \quad e_{max} = \frac{1}{L} \sum_{l=1}^{L} \max_{1 \le i \le |\mathcal{M}_{Q,T}^l|} e_i^l. \quad (15)$$

**Effect of quantization bitwidth.** Table 2 reports $e_{avg}$, $e_{max}$, ACC, and BWT across different bitwidths. To isolate bitwidth effects, we disable QEA projection ($\alpha = 0$) and set the outlier proportion to zero ($p = 0$). As expected, larger bitwidths substantially reduce both error metrics, with accuracy converging to full-precision levels. A sharp performance drop occurs below 5 bits, caused by increased projection distortion and resulting gradient drift.

**Effect of outlier proportion.** In Table 3, we fix the quantization to 4-bit with $\alpha = 0$ while varying $p$. Even with low $e_{avg}$, a single poorly quantized basis vector (large $e_{max}$ at $p = 0$) can destabilize training by inducing a gradient drift. Introducing a small outlier proportion (e.g., $p = 0.5\%$) markedly reduces $e_{max}$ and improves performance relative to the zero-outlier case.

## 4.3 EFFECT OF QEA PROJECTION ON QUANTIZATION ROBUSTNESS

To counteract quantization error accumulation, QGPM integrates Quantization Error-Aware (QEA) projection, which adaptively relaxes orthogonality based on quantization fidelity. We study its effect by varying the error scaling factor $\alpha$, which controls the strength of the orthogonality constraint. Table 4 reports performance as a function of $\alpha$, with the outlier proportion fixed at $p = 1\%$, a setting where QEA has the most pronounced impact. All other hyperparameters match those used in Tables 2 and 3. For each basis vector, the orthogonality weight is $\lambda_i = 1 - \alpha e_i$, where $e_i$ is the quantization error from Eq. 10. We also report the average of orthogonality weight across all basis vectors,

$$\lambda_{avg} = \frac{1}{\sum_{l=1}^{L} |\mathcal{M}_{Q,T}^l|} \sum_{l=1}^{L} \sum_{i=1}^{|\mathcal{M}_{Q,T}^l|} \lambda_i^l. \quad (16)$$

Given a fixed average quantization error (e.g., $e_{avg} = 0.376$ set by the bitwidth and $p$), increasing $\alpha$ lowers $\lambda_{avg}$, thereby relaxing the orthogonality constraint and permitting greater flexibility in gradient updates. This flexibility compensates for subspace distortion: as $\alpha$ rises from zero, accuracy consistently improves, showing that the parallel gradient component offsets quantization error. However, beyond a threshold, performance declines due to over-relaxation, which increases inter-task interference. Thus, $\alpha$ must be calibrated to match the severity of quantization. In practice, larger $\alpha$ values are required under higher quantization error, e.g., Table 9 in Appendix A.3 shows that 4QGPM

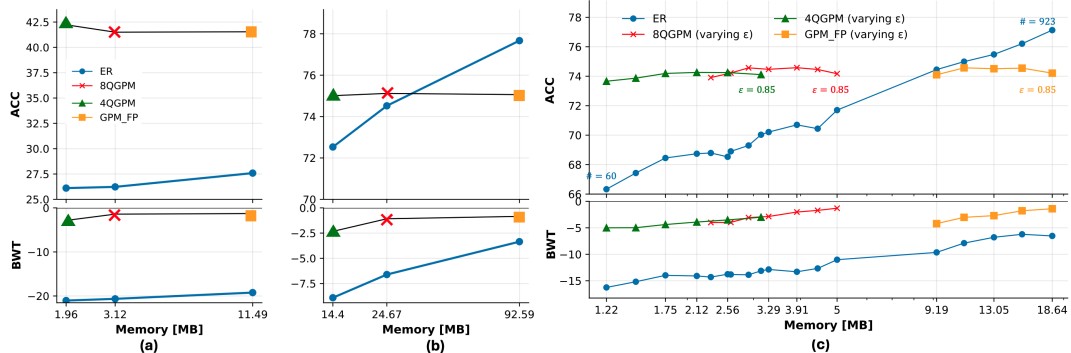

Figure 3: **Memory–accuracy tradeoffs of QGPM. (a,b)** QGPM consistently maintains full-precision level performance, outperforming the ER baseline in the lower-memory regime on ViT-Tiny and ViT-B/16. **(c)** Memory-accuracy curve with varying $\epsilon$ value on ViT-S. QGPM performance remains stable as $\epsilon$ decreases; 4QGPM with a moderate $\epsilon$ achieves the best efficiency, while maintaining full-precision level performance.

generally uses a larger $\alpha$ than 8QGPM. Overall, QEA proves most beneficial under aggressive quantization settings, where error-aware relaxation can significantly stabilize learning.

### 4.4 MEMORY-ACCURACY TRADEOFFS IN QGPM

The memory characteristics of QGPM depend on two factors: (i) the embedding dimension (i.e., model size) and (ii) the threshold $\epsilon$ (i.e., subspace approximation fidelity). We utilize the 10-split ImageNet-R (Hendrycks et al., 2021) benchmark to effectively evaluate models of varying sizes, including ViT-Tiny, ViT-S, and ViT-B/16.

**Effect of model size.** We evaluate QGPM on ViT-Tiny (192-dim) and ViT-B/16 (768-dim) using 10-split ImageNet-R, with ER (Rolnick et al., 2019) as a representative rehearsal baseline (Fig. 3(a) and (b)). GPM memory scales sharply with model size, as its dimensionality matches the embedding dimension of the transformer. For example, fully populated GPMs reach 34MB for ViT-Tiny and 515MB for ViT-B/16. On smaller models, even GPM-FP outperforms ER under equal memory. On larger models, however, ER surpasses GPM-FP at 92.59MB. By contrast, QGPM compresses memory to as little as 14.4MB while still outperforming ER, with negligible accuracy loss. This validates QGPM's effectiveness in scaling to large models under tight memory budgets.

**Effect of threshold $\epsilon$.** Figure 3(c) shows performance as $\epsilon$ decreases from 0.85 in steps of 0.03, evaluated on ViT-S. Smaller $\epsilon$ yields a coarser subspace approximation (fewer bases), reducing QGPM size but potentially increasing forgetting. Performance remains largely stable across a wide range of $\epsilon$, suggesting that memory can be reduced further with modest trade-offs. Notably, 4QGPM with a moderate $\epsilon$ is often superior to 8QGPM or GPM-FP with an aggressively small $\epsilon$.

## 5 CONCLUSION

We introduced QGPM, a scalable and memory-efficient framework for continual learning that integrates: (i) distribution-aware quantization of gradient subspaces, (ii) a Quantization Error-Aware (QEA) projection that adaptively relaxes orthogonality constraints, and (iii) On-the-Fly Sparse Sketching to reduce memory and computation during training. Theoretical analysis and extensive experiments demonstrate that QGPM achieves strong performance under tight memory budgets, consistently outperforming prior methods in both accuracy and efficiency. These results position QGPM as a practical and scalable solution for memory-efficient, privacy-preserving continual learning.

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

# A EXPERIMENTAL DETAILS

## A.1 DATASETS

Tables 5 and 6 summarize the dataset configurations used in the main experiments. To match the memory constraint of QGPM, the number of samples in the replay buffer is adjusted accordingly. Here, buffer size indicates the number of replay images stored in the buffer. For CIFAR-100, the memory overhead of QGPM is 0.81MB, which is equivalent to 264 training samples. Therefore, the buffer size is set to 280. For the 5-Datasets benchmark, we assume all five datasets are equally represented in the buffer. This results in an effective memory usage of 1.7KB per sample. To match QGPM's memory constraint of 7.24MB, a total of 4,259 samples can be stored in the buffer. For miniImageNet, to match the memory overhead of 2.59 and 2.88MB, 128 and 143 samples are used.

Table 5: Dataset statistics of CIFAR100 and miniImageNet.

|  | Split CIFAR-100 | Split-miniImageNet |
|---|---|---|
| num. of tasks | 10 | 20 |
| input size ($C \times H \times W$) | $3 \times 32 \times 32$ | $3 \times 84 \times 84$ |
| # Classes/task | 10 | 10 |
| # Test samples/task | 1,000 | 500 |
| Memory per sample | 3.072KB | 21.168KB |

Table 6: 5-Datasets statistics. MNIST images are replicated across all RGB channels so that each image has 3 channels.

|  | CIFAR-10 | MNIST | SVHN | Fashion-MNIST | notMNIST |
|---|---|---|---|---|---|
| # Classes | 10 | 10 | 10 | 10 | 10 |
| # Training samples | 47,500 | 57,000 | 69,595 | 57,000 | 16,011 |
| # Validation samples | 2,500 | 3,000 | 3,662 | 3,000 | 842 |
| # Test samples | 10,000 | 10,000 | 26,032 | 10,000 | 1,873 |
| Memory per sample | 3.073KB | 0.785KB | 3.073KB | 0.785KB | 0.785KB |

## A.2 MODEL ARCHITECTURES

The model architectures for the 5-layer AlexNet variant and the 20-layer ResNet-18 variant follow those used in Saha et al. (2021). All networks use ReLU activations and a softmax layer combined with cross-entropy loss at the final classification layer. AlexNet and ResNet-18 are trained from scratch.

For ViT, we adopt a ViT-S, a variant of ViT proposed in Dosovitskiy et al. (2021), with a 384-dimensional patch embedding, 6 transformer blocks forming a total of 25 learnable layers and 6 attention heads. The MLP expansion ratio within each attention block is set to 4, and dropout is applied with a probability of 0.1. ViT-S uses pre-trained weights trained on ImageNet-21k. It takes an input image size of $224 \times 224$ with a patch size of 16. Thus, each input image is resized to $224 \times 224$.

Table 7 shows the fully occupied GPM configuration at each backbone and the corresponding full precision memory overhead.

## A.3 HYPERPARAMETER SETTINGS

In this section, we organize hyperparameters used in the experiment in Section 4. In the experiments with AlexNet and ResNet-18, a learning rate of 0.1 is used for the first task and 0.01 is used for all subsequent tasks. For the experiment with AlexNet, 100 training epochs are used. For the ResNet18, 50 training epochs are used. For the ViT-S and ViT-Tiny, a 0.01 learning rate is used with 10 epochs. For the ViT-B/16, a 0.005 learning rate is used with 20 epochs. In all experiments, SGD is used as an optimizer with a 64 mini-batch size.

In the Table 9, method-specific hyperparameters are introduced. $\epsilon_{th}$ is threshold value of k-rank approximation. $\alpha$ is the *QEA scaling factor* for the QEA gradient projection. All experiments were

Table 7: Maximum size of GPM at each layer and the number of GPM parameters for each network.

| Network | Size of maximum $M^l$ | Memory in Bytes |
|---------|----------------------|-----------------|
| AlexNet | $48 \times 48$, $576 \times 576$, $512 \times 512$, $1024 \times 1024$, $2048 \times 2048$ | 22.28MB |
| ResNet18 | $27 \times 27$, $180 \times 180$, $180 \times 180$, $180 \times 180$, $180 \times 180$, $180 \times 180$, $360 \times 360$, $20 \times 20$, $360 \times 360$, $360 \times 360$, $360 \times 360$, $720 \times 720$, $40 \times 40$, $720 \times 720$, $720 \times 720$, $720 \times 720$, $1440 \times 1440$, $80 \times 80$, $1440 \times 1440$, $1440 \times 1440$ | 34.27MB |
| ViT-S | $768 \times 768$, $384 \times 384$, $384 \times 384$, $384 \times 384$, $1536 \times 1536$, $384 \times 384$, $384 \times 384$, $384 \times 384$, $1536 \times 1536$, $384 \times 384$, $384 \times 384$, $384 \times 384$, $1536 \times 1536$, $384 \times 384$, $384 \times 384$, $384 \times 384$, $1536 \times 1536$, $384 \times 384$, $384 \times 384$, $384 \times 384$, $1536 \times 1536$, $384 \times 384$, $384 \times 384$, $384 \times 384$, $1536 \times 1536$ | 66.37MB |
| ViT-B/16 | $768 \times 768$, $768 \times 768$, $768 \times 768$, $768 \times 768$, $3072 \times 3072$, $768 \times 768$, $768 \times 768$, $768 \times 768$, $768 \times 768$, $3072 \times 3072$, $768 \times 768$, $768 \times 768$, $768 \times 768$, $768 \times 768$, $3072 \times 3072$, $768 \times 768$, $768 \times 768$, $768 \times 768$, $3072 \times 3072$, $768 \times 768$, $768 \times 768$, $768 \times 768$, $768 \times 768$, $3072 \times 3072$, $768 \times 768$, $768 \times 768$, $768 \times 768$, $768 \times 768$, $3072 \times 3072$, $768 \times 768$, $768 \times 768$, $768 \times 768$, $768 \times 768$, $3072 \times 3072$, $768 \times 768$, $768 \times 768$, $768 \times 768$, $768 \times 768$, $3072 \times 3072$, $768 \times 768$, $768 \times 768$, $768 \times 768$, $768 \times 768$, $3072 \times 3072$ | 515.25MB |

Table 8: Model parameters and GPM size comparison.

| Network | Category | Memory in Bytes |
|---------|----------|-----------------|
| **AlexNet** | Model Param. size | 26.1MB |
| | GPM_max size | 22.28MB |
| **ResNet-18** | Model Param. size | 4.4MB |
| | GPM_max size | 34.27MB |
| **ViT-S** | Model Param. size | 42.18MB |
| | GPM_max size | 66.3MB |
| **ViT-Tiny** | Model Param. size | 21.15MB |
| | GPM_max size | 34MB |
| **ViT-B/16** | Model Param. size | 327.59MB |
| | GPM_max size | 515MB |

conducted on a single AMD Vega 20 GPU.

## B  GRADIENT PROJECTION MEMORY (GPM)

Table 9: Hyperparameters used for all baselines.

| Methods | Hyperparameters |
|---------|-----------------|
| GPM-FP | $\epsilon_{th}$: 0.9 (10 cifar100), 0.93 (5datasets), 0.93 (miniimagenet) |
| GPM-MC | $\epsilon_{th}$: 0.7 (10 cifar100, 0.79MB), 0.75 (5 datasets, 8.36MB), 0.52 (10 miniimagenet, 2.66MB), 0.51 (20 miniimagenet, 2.88MB) |
| AGEM-MC | Buffer size: 280 (10 cifar100), 4259 (5datasets), 128 (10 miniimagenet), 143(20 miniimagenet) |
| DER++-MC | Buffer size: 280 (10 cifar100), 4259 (5datasets), 128 (10 miniimagenet), 143 (20 miniimagenet) 
 weight for MSE loss on replay logits: 0.1, weight for CE loss on replay samples: 0.5 |
| ER-MC | Buffer size: 280 (10 cifar100), 4259 (5datasets), 128 (10 miniimagenet), 143 (20 miniimagenet) |
| FDR-MC | Buffer size: 280 (10 cifar100), 4259 (5datasets), 128 (10 miniimagenet), 143 (20 miniimagenet) 
 weight for FDR replay loss: 0.6 |
| 8QGPM | $\alpha$: 10 (cifar100), 10 (5datasets), 5 (10 miniimagenet), 5 (20 miniimagenet) 
 outlier: 1% (cifar100), 1% (5datasets), 1% (10 miniimagenet), 1% (20 miniimagenet) |
| 4QGPM | $\alpha$: 20 (cifar100), 20 (5datasets), 15 (10 miniimagenet), 15 (20 miniimagenet) 
 outlier: 3% (cifar100), 3% (5datasets), 2% (10 miniimagenet), 2% (20 miniimagenet) |

---

**Algorithm 2** On-the-Fly Sparse Sketching

---

1: Let $\mathbf{R}$ be the representation matrix and $\mathbf{r}_i$ be the $i$-th representation vector having $n$ dimension. Let's assume that there are $N$ numbers of representation vector and they arrive one by one.
2: $\mathbf{R} \leftarrow \mathbf{0}_{n \times n}$
3: **for** $\forall i \in \{1, 2, \ldots, N\}$ **do**
4: $\quad seed \leftarrow i$
5: $\quad idx \leftarrow Random\_select(\{1, 2, ..., n\}, seed)$
6: $\quad sign \leftarrow Random\_select(\{-1, +1\}, seed)$
7: $\quad \mathbf{R}[:, idx] \leftarrow \mathbf{R}[:, idx] + sign \cdot \mathbf{r}_i$
8: **end for**

---

We now provide a detailed explanation of the Gradient Projection Memory (GPM) scheme Saha et al. (2021). In continual learning, we aim to minimize the average loss over a sequence of tasks; however, without access to earlier data, models tend to forget previous tasks as they learn new ones. GPM combats this catastrophic forgetting by projecting gradients from the new tasks onto the subspace orthogonal to past task representations, without any data replay.

Let $\mathbf{w}_\tau^l \subset \mathbf{w}_\tau$ denote $l$-th layer parameters trained on task $\tau$ and $\mathbf{R}_\tau^l$ be the input representation to layer $l$ at that point. The output activation of layer $l$ after training on task $\tau$ is given by $\mathbf{w}_\tau^l \cdot R_\tau^l$. We would like this activation to remain unchanged even after learning task $\tau + 1$, i.e., $\mathbf{w}_\tau^l \cdot \mathbf{R}_\tau^l = \mathbf{w}_{\tau+1}^l \cdot \mathbf{R}_\tau^l = (\mathbf{w}_\tau^l + \Delta\mathbf{w}^l) \cdot \mathbf{R}_\tau^l$. This requires that the weight update $\Delta\mathbf{w}^l$ satisfies the orthogonality condition $\Delta\mathbf{w}^l \cdot \mathbf{R}_\tau^l = 0$. To enforce this constraint, the learner stores a set of core basis vectors that span the subspace of $\mathbf{R}_\tau^l$ and restricts future gradient updates to lie in the orthogonal complement of this subspace. We denote this memory (i.e., GPM), for layer $l$ after task $\tau$ by $\mathbf{M}_\tau^l$.

Algorithm 3 outlines a continual learning procedure using GPM. For each task $\tau$, the learner: (1) performs projected gradient descent using the existing basis memory $\{\mathbf{M}_{\tau-1}^l\}$ to remove interference (Lines 1–7); (2) gathers the new layer-$l$ activations $\mathbf{R}_\tau^l$ and orthogonally project out components

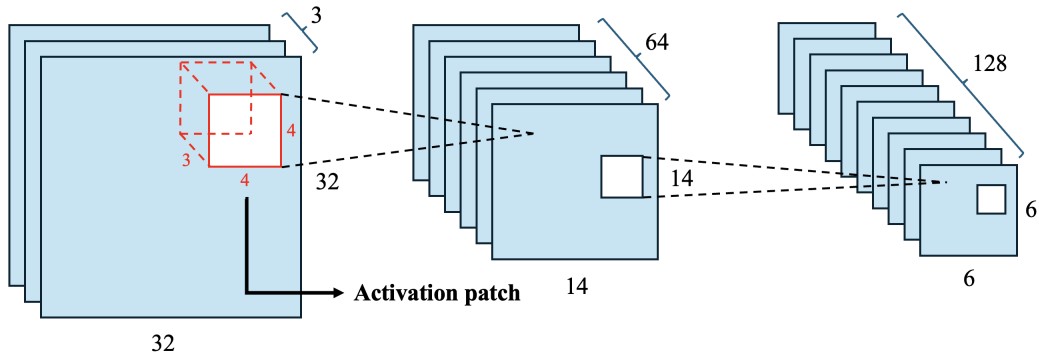

Figure 4: Feature map of AlexNet and its activation patch

---

**Algorithm 3** GPM Algorithm

---

**Input:** $f_{\mathbf{w}}$ the NN model, $\mathcal{D}^{train}$ the training dataset, $\eta$ the learning rate, and $\epsilon_{th}$ the threshold value.
  Initialize, $\mathbf{M}_0^l$, for all $l = 1, 2, \ldots, L'$, and $\mathbf{w} \leftarrow \mathbf{w}_o$.
1: **for** $\tau = 1, 2, \ldots, T$ **do**
2:   **repeat**
3:     $B_n \sim \mathcal{D}_\tau^{train}$
4:     $\mathbf{g} \leftarrow \nabla_{\mathbf{w}} L_\tau$
5:     $\hat{\mathbf{g}} \leftarrow \mathbf{g} - \mathbf{M}_{\tau-1}^l \cdot (\mathbf{M}_{\tau-1}^l)^\top \cdot \mathbf{g}$
6:     $\mathbf{w} \leftarrow \mathbf{w} - \eta \cdot \hat{\mathbf{g}}$
7:   **until** convergence
8:
9:   $B_{n_s} \sim \mathcal{D}_\tau^{train}$
10:   $\{\mathbf{R}_\tau^l\}_{l=1}^{L'} \leftarrow \text{forward}(B_{n_s}, f_{\mathbf{w}})$
11:
12:   **for** $\forall l \in \{1, 2, \ldots, L'\}$ **do**
13:     $\hat{\mathbf{R}}_\tau^l \leftarrow \mathbf{R}_\tau^l - \mathbf{M}_{\tau-1}^l \cdot (\mathbf{M}_{\tau-1}^l)^\top \cdot \mathbf{R}_\tau^l$
14:     $\hat{\mathbf{U}}_\tau^l, \hat{\mathbf{\Sigma}}_\tau^l, \hat{\mathbf{V}}_\tau^l \leftarrow \text{SVD}(\hat{\mathbf{R}}_\tau^l)$
15:     $r \leftarrow \text{criteria}(\hat{\mathbf{R}}_\tau^l, \mathbf{R}_\tau^l, \epsilon_{th})$
16:     $\mathbf{M}_\tau^l \leftarrow \left[\mathbf{M}_{\tau-1}^l, \mathbf{U}_\tau^l[1:r]\right]$
17:   **end for**
18: **end for**

---

already captured by $\mathbf{M}_{\tau-1}^l$ (Line 13); (3) applies SVD to the residual and retains its top singular vectors as the task's core basis (Lines 14–15); (4) concatenates these new bases with $\mathbf{M}_{\tau-1}^l$ to form $\mathbf{M}_\tau^l$ for future tasks (Line 16).

## C ON-THE-FLY SPARSE SKETCHING

### C.1 REPRESENTATION MATIX CONSTRUCTION AND ON-THE-FLY SPARSE SKETCHING

To build the activation (representation) vector $\mathbf{r}_i$ for a convolutional layer, treat each receptive-field patch (activation patch) that will feed the next kernel as one training example: For every image in the mini-batch and every sliding-window position $(u, v)$, we slice the feature map tensor $X \in \mathbb{R}^{C \times H \times W}$ into the cuboid patch $P_{u,v} = X[:, u, u+k, v:v+k]$, where $C_{in}$ is the number of channels and $k$ is the kernel size. By flattening this patch in channel-major order, we obtain a length-$n$ vector where $n = C_{in}k^2$. We generate this flattened feature vector over all mini-batches and the spatial dimension of the feature map. Thus, the total number of feature vectors is $N = s^2 \cdot bsz$, where $s$ is the size of the feature map and $bsz$ is the number of samples in the mini-batch. Thus, if we construct a feature matrix without a sparse sketch, the size of this matrix is $\mathbb{R}^{(k^2 \cdot C_{in}) \times (s^2 \cdot bsz)}$. The resulting representation matrix size for each layer constructed in this way is shown in the Table 10.

Table 10: The size of the representation matrix for each layer without sketching.

| Network | Size of input representation | Memory in Bytes |
|---|---|---|
| AlexNet | $48 \times 20184$, $576 \times 14400$, $512 \times 2500$, $1024 \times 125$, $2048 \times 125$ | 41.46MB |
| ResNet18 | $27 \times 17640$, $180 \times 17640$, $180 \times 17640$, $180 \times 17640$, $180 \times 17640$, $180 \times 4410$, $360 \times 4410$, $20 \times 4410$, $360 \times 4410$, $360 \times 22050$, $360 \times 6050$, $720 \times 6050$, $40 \times 6050$, $720 \times 12100$, $720 \times 12100$, $720 \times 3600$, $1440 \times 3600$, $80 \times 3600$, $1440 \times 3600$, $1440 \times 3600$ | 258.65MB |
| ViT-S | $768 \times 25088$, $384 \times 25216$, $384 \times 25216$, $384 \times 25216$, $1536 \times 25216$, $384 \times 25216$, $384 \times 25216$, $384 \times 25216$, $1536 \times 25216$, $384 \times 25216$, $384 \times 25216$, $384 \times 25216$, $1536 \times 25216$, $384 \times 25216$, $384 \times 25216$, $384 \times 25216$, $1536 \times 25216$, $384 \times 25216$, $384 \times 25216$, $384 \times 25216$, $1536 \times 25216$, $384 \times 25216$, $384 \times 25216$, $384 \times 25216$, $1536 \times 25216$ | 1.62GB |
| ViT-B/16 | $768 \times 25088$, $768 \times 25216$, $768 \times 25216$, $768 \times 25216$, $3072 \times 25216$, $768 \times 25216$, $768 \times 25216$, $768 \times 25216$, $3072 \times 25216$, $768 \times 25216$, $768 \times 25216$, $768 \times 25216$, $3072 \times 25216$, $768 \times 25216$, $768 \times 25216$, $768 \times 25216$, $3072 \times 25216$, $768 \times 25216$, $768 \times 25216$, $768 \times 25216$, $3072 \times 25216$, $768 \times 25216$, $768 \times 25216$, $768 \times 25216$, $3072 \times 25216$, $768 \times 25216$, $768 \times 25216$, $768 \times 25216$, $3072 \times 25216$, $768 \times 25216$, $768 \times 25216$, $768 \times 25216$, $3072 \times 25216$, $768 \times 25216$, $768 \times 25216$, $768 \times 25216$, $3072 \times 25216$, $768 \times 25216$, $768 \times 25216$, $768 \times 25216$, $3072 \times 25216$ | 6.13GB |

Algorithm 2 outlines the On-the-Fly sparse sketching procedure, where $N = s^2 \cdot bsz$ representation vectors arrive sequentially. Each incoming $n$-dimensional vector (where $n = k^2 \cdot C_{in}$) is processed by generating a random Rademacher sign and a target column index uniformly sampled from $\{1, \ldots, n\}$. The signed vector is then accumulated into the corresponding column of the sketch matrix $\mathbf{R}$. Table 11 reports the size of the resulting representation matrix with sparse sketching. For example, in the case of ViT-S, the naive construction of the representation matrix incurs a memory overhead of 1624MB before performing SVD on it. Our method reduces this requirement to 66.37MB, achieving a 24.46$\times$ saving. Note that this is intermediate memory reduction (e.g., GPU RAM usage) and not permanent memory (e.g., Flash memory) reduction.

Figure 5 illustrates RAM usage across the ten-task training sequence on the miniImageNet benchmark using ViT-S. Table 12 reports the total training time, average GPM construction time (in seconds), average Random Access Memory (RAM) usage (in MB), peak RAM usage, and the final performance metrics (ACC and BWT). Without sketching, the average memory overhead is approximately 4533MB. By employing sparse sketching, this can be reduced to 3277.09MB without any degradation in performance. The GPM construction time (which includes the SVD on the representation matrix, $r$-rank approximation, and quantization) can be reduced from 115.06s to 10.60s.

Compared to dense Gaussian random projections, On-the-Fly sparse sketching method offers greater computational and memory efficiency. Each incoming activation vector is processed in $O(\text{nnz}(\mathbf{R}))$ time, using only sign flipping and bucketed additions with two light-weight hash functions, rather than a full $O(nN)$ matrix-vector multiplication. Here, $\text{nnz}(\mathbf{R})$ denotes the number of non-zero columns in $\mathbf{R}$.

Table 11: The size of the representation matrix for each layer with sketching.

| Network | Size of input representation | Memory in Bytes |
|---------|------------------------------|-----------------|
| AlexNet | $48 \times 48$, $576 \times 576$, $512 \times 512$, $1024 \times 125$, $2048 \times 125$ | 3.72MB |
| ResNet18 | $27 \times 27$, $180 \times 180$, $180 \times 180$, $180 \times 180$, $180 \times 180$, $180 \times 180$, $360 \times 360$, $20 \times 20$, $360 \times 360$, $360 \times 360$, $360 \times 360$, $720 \times 720$, $40 \times 40$, $720 \times 720$, $720 \times 720$, $720 \times 720$, $1440 \times 1440$, $80 \times 80$, $1440 \times 1440$, $1440 \times 1440$ | 34.27MB |
| ViT-S | $768 \times 768$, $384 \times 384$, $384 \times 384$, $384 \times 384$, $1536 \times 1536$, $384 \times 384$, $384 \times 384$, $384 \times 384$, $1536 \times 1536$, $384 \times 384$, $384 \times 384$, $384 \times 384$, $1536 \times 1536$, $384 \times 384$, $384 \times 384$, $384 \times 384$, $1536 \times 1536$, $384 \times 384$, $384 \times 384$, $384 \times 384$, $1536 \times 1536$, $384 \times 384$, $384 \times 384$, $384 \times 384$, $1536 \times 1536$ | 66.37MB |
| ViT-B/16 | $768 \times 768$, $768 \times 768$, $768 \times 768$, $768 \times 768$, $3072 \times 3072$, $768 \times 768$, $768 \times 768$, $768 \times 768$, $768 \times 768$, $3072 \times 3072$, $768 \times 768$, $768 \times 768$, $768 \times 768$, $768 \times 768$, $3072 \times 3072$, $768 \times 768$, $768 \times 768$, $768 \times 768$, $768 \times 768$, $3072 \times 3072$, $768 \times 768$, $768 \times 768$, $768 \times 768$, $768 \times 768$, $3072 \times 3072$, $768 \times 768$, $768 \times 768$, $768 \times 768$, $768 \times 768$, $3072 \times 3072$, $768 \times 768$, $768 \times 768$, $768 \times 768$, $768 \times 768$, $3072 \times 3072$, $768 \times 768$, $768 \times 768$, $768 \times 768$, $768 \times 768$, $3072 \times 3072$, $768 \times 768$, $768 \times 768$, $768 \times 768$, $768 \times 768$, $3072 \times 3072$ | 515.25MB |

Empirically and theoretically, sparse sketch maintains $\ell_2$ norms and leading eigen-directions within Johnson–Lindenstrauss-type bounds, offering subspace quality comparable to dense Gaussian projections. Thus, it provides a compelling trade-off between accuracy and efficiency, making it especially well-suited for real-time, resource-constrained, continual learning settings.

Table 12: Resource usage comparison with and without sketching.

| Methods | Without sketching | | With sketching | |
|---------|-------------------|--|----------------|--|
| | ACC (%) | BWT (%) | ACC (%) | BWT (%) |
| 8QGPM | 72.21 | -0.54 | 72.87 | -0.34 |
| GPM construction time [s] | 115.06 | | 10.60 | |
| Total training time [s] | 1768 | | 820 | |
| Average RAM consumption [MB] | 4533.50 MB | | 3277.09 MB | |
| Peak RAM consumption [MB] | 5603.06 MB | | 3586.50 MB | |

# D MIXED-SCHEME QUANTIZATION AND DE-QUANTIZATION

## D.1 MIXED-SCHEME QUANTIZATION

The CINF quantization scheme performs well when the input distribution is approximately Gaussian. However, it is important to note that the input activation to the very first layer (i.e., the raw image patch) typically exhibits a non-Gaussian distribution. As this effect propagates, the activations in the

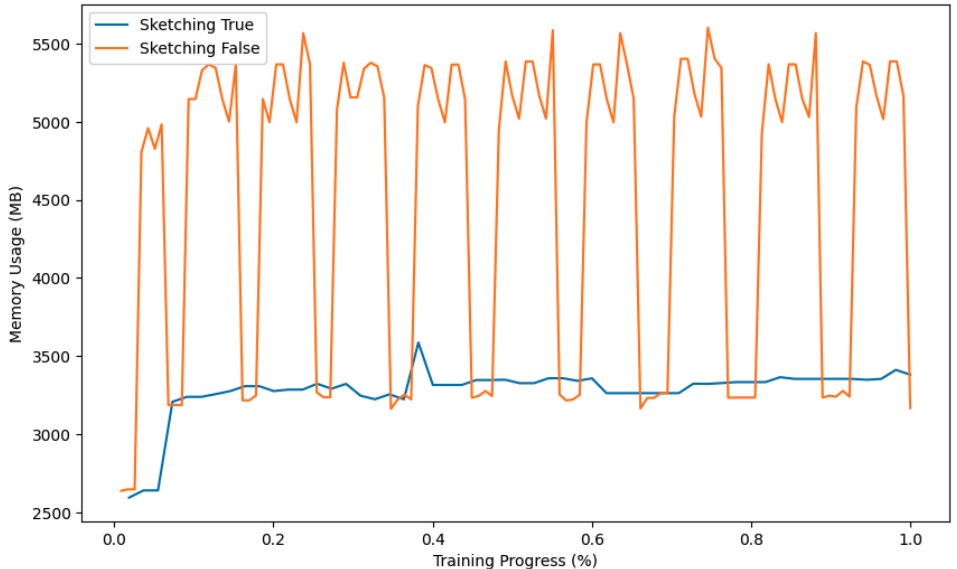

Figure 5: Training-time process RAM usage of 10-split miniImageNet experiment on ViT-S.

early layers also tend to deviate from Gaussianity, particularly during the initial stages of training when model parameters are not yet well-formed. This non-Gaussian behavior can result in significant quantization errors when CINF is applied indiscriminately across all layers. To address this concern, we propose a mixed-scheme quantization strategy where we apply affine quantization in the first layer by default, and conditionally in the second and third layers based on a normality test. The test combines two statistics: kurtosis $\kappa$ of the input vector Mardia (1970) and p-value $p_{\text{SW}}$ from the Shapiro–Wilk Test Shapiro & Wilk (1965). For deeper layers, where representations tend to become more Gaussian due to the central limit effect, we default to CINF quantization. Formally, given an input matrix $\mathbf{U} = [\mathbf{u}_1, \ldots, \mathbf{u}_r]$, the mixed quantization function deployed at the second and third layers can be defined as

$$\text{Quant}(\mathbf{U}[1:r]) = [\text{quant}(\mathbf{u}_j)]_{j=1}^r, \quad \text{quant}(\mathbf{u}) = \begin{cases} \mathcal{Q}_{\text{Affine}}(\mathbf{u}), & \kappa(\mathbf{u}) < 0 \ \wedge \ p_{\text{SW}}(\mathbf{u}) < \varepsilon, \\ \mathcal{Q}_{\text{CINF}}(\mathbf{u}), & \text{otherwise.} \end{cases}$$

$$(17)$$

Definition of *kurtosis* and *p-value* of Shapiro-Wilk Normality test are given as below.

**Definition D.1** *Kurtosis: Kurtosis measures the "tailedness" of the probability distribution. It quantifies whether the data distribution has heavier tails (positive kurtosis) or lighter tails (negative kurtosis) compared to a Gaussian distribution. The kurtosis of a dataset is given by $K = \mathbb{E}[(X - \mu)^4]/\sigma^4 - 3$, where $\mu$ is the mean and $\sigma$ is the standard deviation of the dataset.*

**Definition D.2** *Shapiro-Wilk Test of Normality: The p-value obtained from the Shapiro–Wilk test quantifies the likelihood of the data distribution being Gaussian. Formally, given a null hypothesis $H_0$ (the dataset is Gaussian) and an alternative hypothesis $H_1$ (the dataset is not Gaussian), the p-value measures the probability of observing data at least as extreme as the current dataset under the assumption of normality. Formally, p-value $= P(W \leq W_{obs} \mid H_0)$ where $W$ is the Shapiro–Wilk test statistic. A low p-value (typically less than a threshold $\epsilon$) suggests rejecting $H_0$, indicating that the data significantly deviates from a Gaussian. The statistics can be defined as*

$$W \ = \ \frac{\left(\sum_{i=1}^n a_i \, x_{(i)}\right)^2}{\sum_{i=1}^n \left(x_i - \bar{x}\right)^2},$$

*for a sample of size $n$ with ordered observations $x_{(1)} \leq \cdots \leq x_{(n)}$ and sample mean $\bar{x}$*

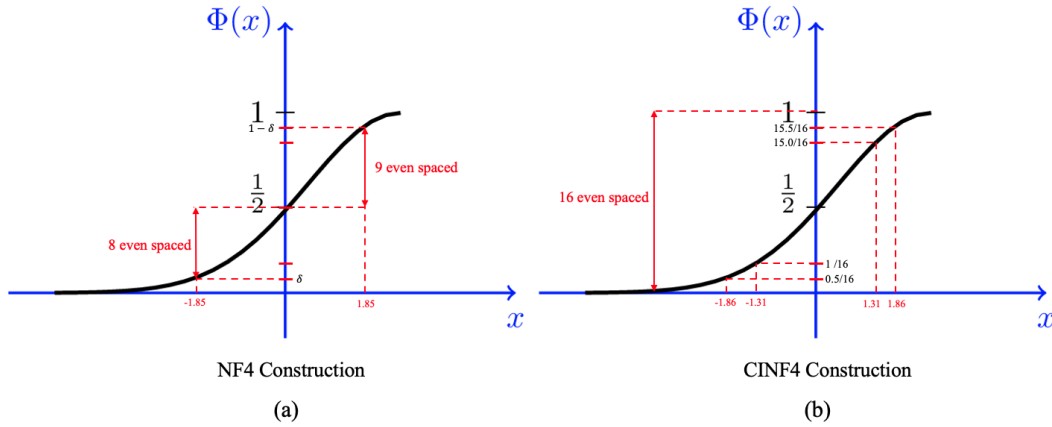

Figure 6: (a) NF4 codebook construction process; (b) 4-bit CINF construction process

**Mixed-scheme dequantization.** In equation equation 11, gradient projection is performed in full precision. Therefore, the quantized GPM must first be dequantized to reconstruct the full-precision projection matrix $\mathbf{P}_{\tau-1}$. Recall that each basis vector stored in the GPM is quantized using either the CINF or Affine scheme. Each scheme requires a distinct set of scalar values for dequantization and projection: $(s, \mu, \lambda_i)$ in the case of CINF, and $(\Delta, z, \lambda)$ in the case of Affine. To handle the mixed quantization setting, QGPM separates the basis vectors into two groups based on their quantization scheme and dequantizes them separately. Formally, for $i \in \{\text{Affine}, \text{CINF}\}$ we find

$$\mathbf{M}^l_{\tau-1,i} = \text{dequant}_i(\mathcal{M}^l_{Q,i}, \mathcal{S}^l_{\tau-1,i}, \mathcal{Z}^l_{\tau-1,i}, \mathcal{O}^l_{\tau-1,i}) \tag{18}$$

$$\mathbf{P}^l_{\tau-1,i} = \mathbf{M}^l_{\tau-1,i} \cdot \mathbf{\Lambda}^l_{\tau-1,i} \cdot (\mathbf{M}^l_{\tau-1,i})^\top, \tag{19}$$

and then compute the final projection matrix as $\mathbf{P}^l_{\tau-1} = \mathbf{P}^l_{\tau-1,\text{Affine}} + \mathbf{P}^l_{\tau-1,\text{CINF}}$.

# E CENTERED INLIER NORMAL FLOAT

## E.1 CODEBOOK CONSTRUCTION

Here, we provide details that distinguish our CINF codebook from NF4. The original NF4 inserts an exact zero value into the codebook to enable lossless quantization of padding or other zero-valued elements. This design choice is appropriate for quantizing model parameters or activations, where exact zeros commonly arise due to operations such as zero-padding.

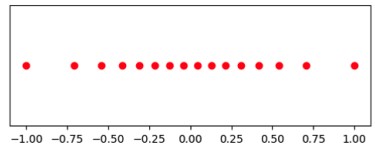

Figure 7: Actual CINF4 codebook.

In contrast, our quantization target is the singular vectors $\hat{\mathbf{U}}^l_\tau[:, i]$ obtained from singular value decomposition (SVD), where such zero-generating operations do not exist. Therefore, we omit the inclusion of an exact zero in the codebook. In the following, we present the codebook construction procedures for both NF4 and CINF4.

**NF4 Codebook construction**

1. Set $\delta = \frac{1}{2}(\frac{1}{32} + \frac{1}{30})$.

2. Compute 8 evenly spaced probability values $p_1, p_2, ..., p_8$, such that $p_1 = \delta, p_8 = 1/2$.

3. Find their corresponding quantile values under Normal Cumulative Density Function (CDF): $\hat{q}_i = \Phi^{-1}(p_i)$ for $i = 0, 1, ..., 8$.

4. Conpute 9 evenly spaced probability values $r_8, r_9, ..., r_{16}$ such that $r_8 = 1/2$ and $r_{16} = 1 - \delta$.

5. Find their corresponding quantile values under Normal CDF: $\hat{q}_i = \Phi^{-1}(r_i)$ for $i = 9, 10, ..., 16$.

6. Normalize the $\hat{q}$ to the range [-1, 1] to get final code: $q_i = \hat{q}_i / \max_i |\hat{q}_i|$.

**CINF4 Codebook construction**

1. Compute 16 evenly spaced probability values $p_1, p_2, \ldots, p_{16}$, such that $p_1 = 15.5/16, p_{16} = 0.5/16$.

2. Find their corresponding quantile values under Normal Cumulative Density Function (CDF): $\hat{q}_i = \Phi^{-1}(p_i)$ for $i = 0, 1, ..., 8$.

3. Normalize the $\hat{q}$ to the range [-1, 1] to get final code: $q_i = \hat{q}_i / \max_i |\hat{q}_i|$.

Figure 6 visualizes the aforementioned process of equal spacing Normal CDF, and Figure 7 shows the actual CINF4 codebook: [-1.000, -0.707, -0.542, -0.416, -0.310, -0.215, -0.127, -0.042, 0.042, 0.127, 0.215, 0.310, 0.416, 0.542, 0.707, 1.000].

# F   ADDITIONAL EXPERIMENTS

## F.1   EXPERIMENTS WITH EMPTY VIT

Table 13: ACC and BWT for 8QGPM and baselines on the empty ViT model

| Methods | 10-split miniImageNet | |
|---|---|---|
| | ACC (%) | BWT (%) |
| Memory | 1.87 MB | |
| GPM-FP | 48.47 | -3.49 |
| Memory | 0.52 MB | |
| GPM-MC | 40.64 | -15.23 |
| AGEM-MC | 23.07 | -37.00 |
| DER++-MC | 26.84 | -33.24 |
| ER-MC | 25.02 | -34.64 |
| FDR-MC | 24.36 | -34.04 |
| PCAOGD-MC[†] | 20.47 | -34.48 |
| **8QGPM (Ours)** | **48.39** | **-3.59** |

In the main paper, we use ViT-S with a pre-trained weight. Typically, ViT lacks strong inductive biases such as CNN, which makes them less data efficient and harder to train from scratch. In the case of the continual learning scenario, we divide the dataset into several chunks to make a task. Thus, the amount of each task is extremely insufficient to train the ViT model from scratch. In this section, we investigate the continual learning performance of QGPM over other baselines on the empty ViT model, which is trained from scratch. For the experiment, we use a custom ViT model, which has 128 patch embedding, 6 transformer blocks, 4 multi-head attention with 4 MLP expansion ratio, forming 25 learnable layers. Model parameter size is 4.7MB, and fully occupied GPM size is 7.2MB.

The results of Table 13 validate that QGPM maintains full-precision-level performance even when ViT is trained from scratch. Notably, QGPM incorporates fewer basis vectors in this setting, resulting in a smaller memory footprint compared to the pretrained ViT case in the main paper. As a result, the rehearsal-based baseline struggles significantly due to the severely constrained replay buffer.

## F.2 EXPERIMENTS WITH OTHER DATASETS ON RESNET18

Table 14: The results of continual learning with QGPM on miniImageNet using ResNet18.

| Methods | 10-split CIFAR100 | | 10-split miniImageNet | |
|---|---|---|---|---|
| | ACC (%) | BWT | ACC (%) | BWT |
| SGD | 54.74 | -26.79 | 33.21 | -25.92 |
| GPM-FP | 70.23 | -1.09 | 53.51 | -1.61 |
| Memory | 60.32MB | | 22.94MB | |
| **8QGPM (Ours)** | **70.46** | **-1.24** | **52.83** | **-1.93** |
| Memory | 5.41MB | | 5.99MB | |

Table 14 presents the experimental results on 10-split CIFAR-100 and 10-split miniImageNet using ResNet-18, as reported in the main paper. As shown, QGPM consistently maintains near full-precision performance on various datasets on ResNet18.

## F.3 EXPERIMENTS ON AN NLP TASK

Table 15: Continual learning result of QGPM with text classification task.

| Bit-width | SGD | | Full Precision | | 8-bit QGPM | |
|---|---|---|---|---|---|---|
| | ACC (%) | BWT (%) | ACC (%) | BWT (%) | ACC (%) | BWT (%) |
| Performance | 75.73 | -10.38 | 77.80 | -6.15 | 77.97 | -5.97 |
| Memory | – | | 266KB | | 75KB | |

While the main paper primarily focuses on vision tasks, in this section, we demonstrate that QGPM can be extended to other domains, such as natural language processing (NLP). We use the *Yahoo Answers Topics* dataset for a topic classification task. Each sample consists of a *question title* (a short string summarizing the question) and *question content*, which provides a longer, more detailed description. The label is an integer range from 0 to 9, representing one of ten coarse-grained categories (e.g., Health, Sports, etc.). The dataset contains 1,400,000 training samples and 60,000 test samples. While text lengths vary across samples, they average around 300 words. For the continual learning setup, we split the ten classes into five tasks, each containing two classes.

We use a transformer-based lightweight encoder for text classification. It consists of:

1. Token embedding of dimension 128

2. Sinusoidal positional encoding

3. 256 MLP hidden dimension

4. 2 transformer block with 4 multi-head self-attention module

5. Redisual connections around both the attention and MLP layers with LayerNorm

The encoder consists of a total of 7 learnable layers, and the input activation before each layer is extracted. The fully occupied GPM size and the representation matrix size without sketching for this network are shown in Tables 16 and 17, respectively.

We train the encoder using stochastic gradient descent (SGD) with learning rate scheduling. Across all experiments, the mini-batch size, number of training epochs, and initial learning rate are fixed to 64, 20, and 0.02, respectively. For the GPM parameters, the threshold value $\epsilon_{th}$ is set to 0.93. For 8-bit quantization, QEA scaling factor $\alpha$ and outlier percentage $p$ are set to 10 and 2, respectively; for 4-bit quantization, they are set to 20 and 3.

Table 15 illustrates that QGPM achieves performance comparable to the full-precision baseline, while significantly reducing memory overhead. As shown in Table 16, the fully occupied GPM-FP has a size of 812 KB, which can be reduced to 228 KB and 143 KB using 8-bit and 4-bit quantization, respectively.

| Network | Size of maximum $M^l$ | Memory in Bytes |
|---|---|---|
| Encoder (7 layers) | $128 \times 128$, $128 \times 128$, $128 \times 128$, $256 \times 256$, $128 \times 128$, $128 \times 128$, $256 \times 256$ | 0.812MB |

Table 16: The maximum GPM size of the Transformer Encoder (Appendix F.3)

| Network | Size of input representation | Memory in Bytes |
|---|---|---|
| Encoder (7 layers) | $128 \times 10600$, $128 \times 10600$, $128 \times 10600$, $256 \times 10600$, $128 \times 10600$, $128 \times 10600$, $256 \times 10600$ | 46.5MB |

Table 17: The representation matrix size of the Transformer Encoder (Appendix F.3) without sparse sketching

## F.4 REHEARSAL BUFFER ADJUSTMENT STRATEGY

To ensure a fair comparison under identical memory constraints, we adjusted the number of rehearsal samples stored in the buffer for each rehearsal-based method (e.g., GEM-MC, DER-MC, FDR-MC, ER-MC) so that their total memory usage matched that of QGPM. We computed the per-sample memory footprint (i.e., image resolution $\times$ number of channels $\times$ bit depth) and reduced the number of stored samples accordingly — for example, constraining to 0.8MB in the CIFAR-100 setup. This strategy aligns with prior works on memory-constrained continual learning Zhou et al. (2023); Iscen et al. (2020) and allows for fair and controlled evaluation.

Table 18 compares two strategies for reducing replay buffer memory: lowering image resolution vs. reducing sample count. While both methods perform similarly compared to the performance of 8QGPM for small images like CIFAR100, resolution reduction degrades performance on larger images like miniImageNet due to severe distribution shifts induced by quantization.

As shown in Table 19, even with more samples (280 and 560) on this resolution-based scheme, the performance worsens, leading to the conclusion that resolution-based quantization is impractical – even for moderately sized images (e.g., miniImageNet).

Table 18: Comparison of memory and performance across buffer strategies.

| Dataset / Model | Metric | DER++ (Full res / Reduced #) | DER++ (Reduced res / Full #) | 8QGPM |
|---|---|---|---|---|
| | Per image mem. size [B] | $32 \times 32 \times 3 \times 1B = 3{,}072$ | $32 \times 32 \times 3 \times 0.5B = 1{,}536$ | – |
| | # in buffer | 280 | $280 \times 2 = 560$ | – |
| CIFAR100 / AlexNet | Buffer mem. size [MB] | 0.8203125 | 0.8203125 | 0.81 |
| | ACC | 60.68 | 65.29 | 70.7 |
| | BWT | $-18.5$ | $-13.2$ | $-0.81$ |
| | Per image mem. size [B] | $84 \times 84 \times 3 \times 1B = 21{,}168$ | $84 \times 84 \times 3 \times 0.5B = 10{,}584$ | – |
| | # in buffer | 93 | $93 \times 2 = 186$ | – |
| miniImageNet / ViT | Buffer mem. size [MB] | 1.968624 | 1.968624 | 0.81 |
| | ACC | 30.01 | 26.23 | 48.39 |
| | BWT | $-26.7$ | $-34.06$ | $-3.59$ |

Table 19: Buffer comparison under different per-image precisions (miniImageNet / ViT).

| Metric | 8-bit (21 168 B) | 4-bit (10 584 B) |
|---|---|---|
| Images in buffer | 280 | 560 |
| Buffer size [MB] | 5.92 | 5.92 |
| Acc [%] | 34.10 | 26.06 |
| BWT [%] | $-21.96$ | $-34.10$ |

# G ADDITIONAL PROOFS

## G.1 PROOF OF THEOREM 3.2

***Quantization Error Accumulation***: *Let $M_o = [\mathbf{v}_1, \mathbf{v}_2, ..., \mathbf{v}_m] \in \mathbb{R}^{n \times m}$ denote a full precision GPM, and $E = [\epsilon_1, \epsilon_2, ..., \epsilon_m] \in \mathbb{R}^{n \times m}$ be a Gaussian error matrix with each column $\epsilon_j \in \mathbb{R}^n$ is drawn from $\mathcal{N}(0, \sigma^2 I_n)$. We define the quantized GPM incorporating error as $M_e = M_o + E = [\mathbf{v}_1 + \epsilon_1, \mathbf{v}_2 + \epsilon_2, ..., \mathbf{v}_m + \epsilon_m] \in \mathbb{R}^{n \times m}$. When naive SGD produces a gradient $\mathbf{g} \in \mathbb{R}^n$, its orthogonal components with respect to $M_o$ and $M_e$ are denoted by $\hat{\mathbf{g}}_o$ and $\hat{\mathbf{g}}_e$, respectively. Then, it holds that $\mathbb{E}[\|\hat{\mathbf{g}}_o - \hat{\mathbf{g}}_e\|] \geq \|\mathbb{E}[\hat{\mathbf{g}}_o - \hat{\mathbf{g}}_e]\| = m \cdot \sigma^2 \cdot \|\mathbf{g}\|$, implying that the quantization error increases proportionally to the number of error-incorporating basis vector in the GPM and quadratically with $\sigma$ which quantifies the degree of error.*

***Proof:*** Let us define projection matrices for $M_o$ and $M_e$ as

$$P_o = M_o \cdot M_o^T,$$
$$P_e = M_e \cdot M_e^T.$$

The orthogonal component of the gradient $g$ with respect to the subspace spanned by $M_o$ and $M_e$ is found as

$$\hat{\mathbf{g}}_o = (I - P_o) \cdot \mathbf{g},$$
$$\hat{\mathbf{g}}_e = (I - P_e) \cdot \mathbf{g}.$$

Hence, the difference between the projected components is

$$\begin{aligned}
\hat{\mathbf{g}}_o - \hat{\mathbf{g}}_e &= [(I - P_o) - (I - P_e)]\mathbf{g} \\
&= (P_o - P_e)\mathbf{g} \\
&= (M_o \cdot M_o^T - M_e \cdot M_e^T)\mathbf{g} \\
&= (M_o \cdot M_o^T - (M_o + E) \cdot (M_o + E)^T)\mathbf{g} \\
&= -(M_o E^T + E M_o^T + E E^T)\mathbf{g}.
\end{aligned}$$

Since the error matrix $E$ is Gaussian distributed with each column $e_j \in R^n$ drawn from $N(0, \sigma^2 I_n)$,

$$\mathbb{E}[(M_o E^T)\mathbf{g}] = M_o \cdot \mathbf{g} \cdot \mathbb{E}[E^T] = 0$$
$$\mathbb{E}[(E M_o^T)\mathbf{g}] = M_o^T \cdot \mathbf{g} \cdot \mathbb{E}[E] = 0$$
$$\mathbb{E}[E \cdot E^T] = \mathbb{E}[\sum_{j=1}^m e_j \cdot e_j^T] = \sum_{j=1}^m \mathbb{E}[e_j \cdot e_j^T] = m \cdot \sigma^2 \cdot I$$

Thus, the expectation of $\hat{\mathbf{g}}_e - \hat{\mathbf{g}}_o$ can be found as

$$\mathbb{E}[\hat{\mathbf{g}}_o - \hat{\mathbf{g}}_e] = -m \cdot \sigma^2 \cdot \mathbf{g}.$$

Since $\|\mathbb{E}[X]\| \leq \mathbb{E}[\|X\|]$,

$$\mathbb{E}[\|\hat{\mathbf{g}}_o - \hat{\mathbf{g}}_e\|_2] \geq \|\mathbb{E}[\hat{\mathbf{g}}_o - \hat{\mathbf{g}}_e]\|_2 = m \cdot \sigma^2 \cdot \|\mathbf{g}\|_2.$$

$\square$

## G.2 PROOF OF THEOREM 3.3

$(1 \pm \epsilon)$ $\ell_2$ ***Subspace embedding***: *A matrix $\mathbf{S} \in \mathbb{R}^{r \times k}$ is said to be a $(1 \pm \epsilon)$-subspace embedding for the row space of $\mathbf{A} \in \mathbb{R}^{k \times n}$ in the $\ell_2$-norm if, for all $\mathbf{x} \in \mathbb{R}^n$,*

$$(1 - \epsilon)\|\mathbf{A}\mathbf{x}\|_2^2 \leq \|\mathbf{S}\mathbf{A}\mathbf{x}\|_2^2 \leq (1 + \epsilon)\|\mathbf{A}\mathbf{x}\|_2^2.$$

***Subspace embedding of Sparse Sketch***: *Let $S$ be the $N$ by $r$ sparse-embedding matrix constructed by $h : [N] \rightarrow [r]$ and $\sigma : [N] \rightarrow \{-1, 1\}$ be hash functions. Then the $i$-th column of the sparse-embedding matrix $S$ is non-zero only in the $h(i)$-th row and this non-zero entry has a value*

of $\sigma(i)$. Then with probability $1 - \delta$ for any rank $k$ matrix $A$ and $r = O(\frac{k^2}{\epsilon^2\delta})$, $S$ is a $(1 \pm \epsilon) - l_2$ subspace embedding for the row of $A$.

Before proving the theorem, we state a few definitions and a helpful lemma.

***Definition 1:*** *We say $\mathbf{C}$ is an $\epsilon$-approximate matrix product of $\mathbf{A}$ and $\mathbf{B}$ if it satisfies*

$$\|\mathbf{A}^\top\mathbf{B} - \mathbf{C}\|_F \leq \epsilon\|\mathbf{A}\|_F\|\mathbf{B}\|_F$$

***Definition 2:*** *A distribution $\mathcal{D}$ on $\mathbf{S} \in \mathbb{R}^{N \times r}$ is said to satisfy the $(\epsilon, \delta, l) - JL$ moment property if $\forall \mathbf{x} \in \mathcal{R}^N$ where $\|\mathbf{x}\|_2 = 1$, $\mathbb{E}[|\|\mathbf{S}^T\mathbf{x}\|_2^2 - 1|^l] \leq \epsilon^l\delta$.*

***Definition 3:*** *For a scalar random variable $X$, let $\|X\|_p := \mathbb{E}[|X|^p]^{1/p}$, which is $L^p$ norm on the space $L^p(\Omega)$. Minkowski's inequality gives the triangle inequality: $\|X + Y\|_p \leq \|X\|_p + \|Y\|_p$.*

***Lemma 1*** *Let $l \geq 2, \epsilon, \delta \in (0, 1/2)$, and $\mathcal{D}$ on $\mathbf{S}$ be a distribution that satisfies the $(\epsilon, \delta, l) - JL$ moment property. Then for every pair of matrices $\mathbf{A}, \mathbf{B}$ with $N$ columns,*

$$Pr_{\mathbf{S} \sim \mathcal{D}}\left[\|\mathbf{A}\mathbf{S}\mathbf{S}^\top\mathbf{B}^\top - \mathbf{A}\mathbf{B}^\top\|_F > 3\epsilon\|\mathbf{A}\|_F\|\mathbf{B}\|_F\right] \leq \delta$$

*Proof of Lemma 1:* We first note that for $\mathbf{x}, \mathbf{y} \in \mathbb{R}^d$, $\langle\mathbf{S}^T\mathbf{x}, \mathbf{S}^T\mathbf{y}\rangle = \frac{1}{2}\left(\|\mathbf{S}^T\mathbf{x}\|_2^2 + \|\mathbf{S}^T\mathbf{y}\|_2^2 - \|\mathbf{S}^T(\mathbf{x} - \mathbf{y})\|_2^2\right)$. Thus,

$$\mathbb{E}[\|\langle\mathbf{S}^\top\mathbf{x}, \mathbf{S}^\top\mathbf{y}\rangle - \langle\mathbf{x}, \mathbf{y}\rangle\|_l] = \frac{1}{2}\|(\|\mathbf{S}^\top\mathbf{x}\|_2^2 - \mathbf{x}_2^2) + (\|\mathbf{S}^\top\mathbf{y}\|_2^2 - \mathbf{y}_2^2) - (\|\mathbf{S}^\top(\mathbf{x} - \mathbf{y})\|_2^2 - \|\mathbf{x} - \mathbf{y}\|_2^2)\|_l$$

$$\leq \frac{1}{2}\left(\|\|\mathbf{S}^\top\mathbf{x}\|_2^2 - \mathbf{x}_2^2\|_l + \|\|\mathbf{S}^\top\mathbf{y}\|_2^2 - \mathbf{y}_2^2\|_l + \|\|\mathbf{S}^\top(\mathbf{x} - \mathbf{y})\|_2^2 - \|\mathbf{x} - \mathbf{y}\|_2^2\|_l\right)$$

$$\leq \frac{1}{2}\left(\epsilon\delta^{1/l} + \epsilon\delta^{1/l} + 4\epsilon\delta^{1/l}\right) \leq 3\epsilon\delta^{1/l}$$

where we first apply the triangle inequality and then apply the JL moment property. From this, we can conclude that for arbitrary $\mathbf{x}, \mathbf{y}$, $\mathbb{E}[\|\langle\mathbf{S}^\top\mathbf{x}, \mathbf{S}^\top\mathbf{y}\rangle - \langle\mathbf{x}, \mathbf{y}\rangle\|_l] \leq 3\epsilon\delta^{1/l}\|\mathbf{x}\|_2\|\mathbf{y}\|_2$ Now since the $ij$-th entry of $\mathbf{A}\mathbf{B}^\top$ is given by $\langle\mathbf{A}^i, \mathbf{B}^j\rangle$, the inner product of the $i$-th column of $\mathbf{A}$ and the $j$-th column of $\mathbf{B}$, we have that

$$\mathbb{E}[\|\mathbf{A}\mathbf{S}\mathbf{S}^\top\mathbf{B}^\top - \mathbf{A}\mathbf{B}^\top\|_F^l] \leq (3\epsilon)^l\delta\sum_{ij}\|\mathbf{A}^i\|_2^l\|\mathbf{B}^j\|_2^l \leq (3\epsilon)^l\delta(\sum_{ij}\|\mathbf{A}^i\|_2^2\|\mathbf{B}^j\|_2^2)^{l/2}$$

$$= (3\epsilon)^l\delta(\|\mathbf{A}\|_F^2\|\mathbf{B}\|_F^2)^{l/2}$$

$$= (3\epsilon)^l\delta(\|\mathbf{A}\|_F^l\|\mathbf{B}\|_F^l)$$

where the first line follows from the triangle inequality, and the second inequality is from plugging in the inequality derived previously. Now we plug this into Markov's inequality to get that

$$Pr[\|\mathbf{A}\mathbf{S}\mathbf{S}^\top\mathbf{B}^\top - \mathbf{A}\mathbf{B}^\top\|_F^l > (3\epsilon)^l\|\mathbf{A}\|_F^l\|\mathbf{B}\|_F^l] \leq \frac{\mathbb{E}[\|\mathbf{A}\mathbf{S}\mathbf{S}^\top\mathbf{B}^\top - \mathbf{A}\mathbf{B}^\top\|_F^l]}{(3\epsilon)^l\|\mathbf{A}\|_F^l\|\mathbf{B}\|_F^l} \leq \delta$$

This will be used to prove the final theorem.

**Proof of the theorem:** We want to show that if $\mathbf{S}$ is the sparse embedding matrix with at least $r = \frac{2}{\epsilon^2\delta}$ rows, $\mathbf{S}$ satisfies the $(\epsilon, \delta, 2) - JL$ moment property. We need to show that for a unit vector $\mathbf{x}$ with $\|\mathbf{x}\|_2 = 1$, $\mathbb{E}[(\|\mathbf{S}^\top\mathbf{x}\|_2^2 - 1)^2] \leq \epsilon^2\delta$. We do this by expanding to get $\mathbb{E}[\|\mathbf{S}^\top\mathbf{x}\|_2^4 - 2\mathbb{E}[\|\mathbf{S}^\top\mathbf{x}\|_2^2]] + 1$; the middle term is 1 and from expansion we can show that $\mathbb{E}[\|\mathbf{S}^\top\mathbf{x}\|_2^4 \leq 1 + \frac{2}{r}$, so $\mathbb{E}[(\|\mathbf{S}^\top\mathbf{x}\|_2^2 - 1)^2] \leq \frac{2}{r}$. Thus if $r > \frac{2}{\epsilon^2\delta}$, the $(\epsilon, \delta, 2) - JL$ moment property hold. Let $\mathbf{V}$ be an orthonormal basis for the rows of $\mathbf{A}$. Now since $\mathbf{S}$ satisfies the $(\epsilon, \delta, 2) - JL$ moment property.

$$Pr\left[\mathbf{V}^\top\mathbf{S}\mathbf{S}^\top\mathbf{V} - \mathbf{V}^\top\mathbf{V} > 3\epsilon\|\mathbf{V}\|_F\|\mathbf{V}\|_F\right] \leq \delta$$

$$\implies Pr\left[\|\mathbf{V}^\top\mathbf{S}\mathbf{S}^\top\mathbf{V} - \mathbf{I}_k\|_F > 3\epsilon k\right] \leq \delta$$

so, with $\epsilon = \frac{\epsilon'}{k}$, we get $r = \mathcal{O}\left(\frac{k^2}{\epsilon'^2\delta}\right)$ columns needed. $\qquad\square$

