# OpenReview forum: "Quantized Gradient Projection for Memory-Efficient Continual Learning"
_ICLR.cc/2026/Conference — ICLR 2026 Poster_

### Official Review · Reviewer_8rDA · 2025-10-27

**Soundness:** 2
**Presentation:** 3
**Contribution:** 2
**Rating:** 4
**Confidence:** 4

**Summary:**

Summary and Strengths
This paper presents Quantized Gradient Projection Memory (QGPM) framework that combines basis-wise quantization, quantization-error-aware projection, and sparse sketching for memory-efficient continual learning.

**Strengths:**

The core idea of applying quantization to gradient subspace storage is an interesting and novel direction, as it tackles the important issue of memory accumulation in projection-based continual learning methods. The authors’ empirical evaluation is thorough, spanning several benchmarks (CIFAR-100, miniImageNet, 5-Datasets) and architectures (AlexNet, ResNet-18, ViT). The proposed method consistently achieves competitive accuracy under stringent memory budgets, demonstrating that quantizing for gradient projection can indeed mitigate forgetting while reducing storage costs.
Another appealing design choice is to retain overloaded (outlier) values in high precision instead of truncating them, which differs from conventional quantization and appears to contribute to robustness  (though, in my view, this intriguing design deserves deeper discussion). Overall, the paper is clearly written and includes extensive experiments supporting the empirical claims.

**Weaknesses:**

- Simplistic treatment of quantization - While the work proposes a quantization method, its treatment of quantization theory remains rather naive. The paper frequently refers to the scheme as “information-theoretically optimal,” yet the design is essentially a variant of scalar quantization. From rate–distortion theory, vector quantization is provably superior to scalar methods. There is little justification for why this scalar approach is adopted, and the omission of stochastic or vector quantization (or even a discussion thereof) weakens the claim of theoretical optimality.

- Limited insight from the quantization analysis -  The theoretical analysis (e.g., Theorem 3.1) focuses on quantizing an i.i.d. Gaussian source. However, the quantization of Gaussian sources is a well-understood topic, and textbook references (e.g., Lecture Notes on Information Theory by Polyanski and Wu) describe optimal scalar and vector quantizers for such distributions. As such, the purpose of the analysis and the insights drawn from it are not very clear. Clarifying the purpose and novelty of this analysis would improve the paper.

- Disconnect between theoretical analysis and key continual-learning metrics -  A large portion of the paper and appendices is devoted to theoretical derivations related to quantization distortion, yet the core continual-learning metrics, namely, learning performance and memory usage, are analyzed only empirically. Theoretical discussion does not connect quantization distortion to these global measures. If such an analysis is intractable, the paper should explain why; otherwise, incorporating at least approximate analytical links between quantization distortion and forgetting performance would significantly strengthen the contribution.

- Restricted and oversimplified neural network assumptions - Section 3.1 appears to base the analysis on linear layers without bias terms, which limits the generality of the conclusions. In realistic neural networks, nonlinear activations such as ReLU clip all negative activations to zero, fundamentally changing the distribution and thus the quantization distortion behavior. The authors should discuss how such nonlinearities might alter or invalidate the derived results and whether the approach extends beyond this simplified setting.



Overall,  the paper explores an appealing and practically relevant idea of quantizing gradient subspaces for memory-efficient continual learning, and backs it with solid experiments. However, the theoretical claims are overstated, and the quantization analysis lacks depth and connection to learning outcomes. Addressing these conceptual limitations, clarifying the notion of “information-theoretic optimality,” and broadening the analysis to more realistic neural architectures would be essential for the work to meet the standards of ICLR.

**Questions:**

See above under weaknesses

---

> ### Author Response · Authors · 2025-11-25
>
> **[Weakness 1]** "Simplistic treatment of quantization - While the work proposes a quantization method, its treatment of quantization theory remains rather naive. The paper frequently refers to the scheme as 'information-theoretically optimal,' yet the design is essentially a variant of scalar quantization. From rate–distortion theory, vector quantization is provably superior to scalar methods. There is little justification for why this scalar approach is adopted, and the omission of stochastic or vector quantization (or even a discussion thereof) weakens the claim of theoretical optimality."
>
> ---
>
> We thank the reviewer for this point, which helps us improve the clarity of our presentation. Our use of the phrase “information-theoretically optimal” may have indeed unintentionally suggested that we claimed scalar quantization to be optimal in general – this is certainly not the case and we appreciate the opportunity to clarify.
>
> Our intent here was to convey that we rely on the fact that quantile-based scalar quantization is information-theoretically optimal within the class of scalar quantizers under Gaussian distributions. This is well-established in scalar rate-distortion theory, where making the quantile bins uniform minimizes reconstruction error under a fixed number of quantization levels. Building on this, CINF approximates quantile quantization by matching the GPM basis vector statistics to a distribution of fixed codebook derived from Gaussian. This strategy, inspired by NF4 quantization, is not itself "optimal," but aims to preserve the distribution-aware design that is at the core of scalar quantization's efficiency.
>
> To clarify this, we now cite [1], which rigorously analyzes NF4 and shows that while it is not rate-distortion optimal in the full sense, it performs as a robust and efficient approximation to quantile quantization for Gaussian data. We will update our paper to better reflect this distinction and clarify theoretical claims.
>
> **Regarding the omission of vector quantization (VQ)**: this is a deliberate choice, motivated by both theoretical and practical constraints. In our setting, the object being quantized is not generic data, but orthonormal basis vectors of the GPM, obtained via SVD. These vectors are unit-norm, mutually orthogonal, and needed for interference-free projection.
>
> Vector quantization benefits from reusing the same codeword for similar data, even if that means introducing some distortion. If two carefully designed orthogonal basis vectors are mapped to the same codeword, they collapse to a single direction – basically, a dimension of the GPM would collapse, defeating the purpose of gradient projection memory. To avoid this, we would have to give each basis vector its own unique codeword, which would cancel out the memory savings that quantization is supposed to provide.
>
> Additionally, unlike classical rate-distortion settings which assume that all quantization targets are available offline, QGPM operates in a continual learning regime, where new basis vectors are produced sequentially as new tasks arrive in online fasion. Applying VQ here would require either retraining the codebook online, which would change old codewords, or freezing the codebook early, which leads to poor adaptation to future tasks. In contrast, scalar quantization readily allows per-vector centering and scaling, ultimately providing low-error compression without codeword drift.
>
> For the above reasons, we restrict ourselves to scalar quantization of basis vectors in QGPM. We will clarify this choice and include this discussion in the revised version of the manuscript.
>
> [1] Davis Yoshida. NF4 isn’t information theoretically optimal (and that’s good). CoRR, abs/2306.06965, 2023.

---

> ### Author Response · Authors · 2025-11-25
>
> **[Weakness 2]** "Limited insight from the quantization analysis - The theoretical analysis (e.g., Theorem 3.1) focuses on quantizing an i.i.d. Gaussian source. However, the quantization of Gaussian sources is a well-understood topic, and textbook references (e.g., Lecture Notes on Information Theory by Polyanski and Wu) describe optimal scalar and vector quantizers for such distributions. As such, the purpose of the analysis and the insights drawn from it are not very clear. Clarifying the purpose and novelty of this analysis would improve the paper."
>
> ---
>
> We agree that the quantization of i.i.d. Gaussian sources is a well-established topic in information theory; both optimal scalar quantization (e.g., via Lloyd–Max iterations) and vector quantization strategies for such distributions can be found in classical references. However, our theoretical analysis serves a different purpose.
>
> Our goal is not to contribute new rate-distortion results or re-derive known quantizers. Instead, our analysis aims to identify a practical failure mode of a specific static quantization strategy, the max-based normalization used in NFk quantizers. This failure becomes especially pronounced when applied to high-dimensional orthonormal basis vectors obtained through singular value decomposition. Theorem 3.1 relies on classical order-statistics results [2] to argue why such max-based schemes become increasingly ill-conditioned as the basis dimension grows: the maximum absolute value among standard normal samples increases with dimension, while the central mass remains tightly concentrated. This causes the range of values to become too wide, which means that the middle bins of the quantizer get underused, reducing the overall quantization fidelity. Motivated by this, our CINF avoids normalizing to the global max and instead uses bounded inlier quantiles to match the effective support of the GPM basis vectors to the standard normal codebook. While this is a simple idea, we believe the analysis helps understand the failure mode of the max-based normalization schemes in this setting. We will revise the paper to make this purpose more clear.
>
> [2] H Leon Harter. Expected values of normal order statistics. Biometrika, 48(1/2):151–165, 1961.

---

> ### Author Response · Authors · 2025-11-25
>
> **[Weakness 3]** "Disconnect between theoretical analysis and key continual-learning metrics - A large portion of the paper and appendices is devoted to theoretical derivations related to quantization distortion, yet the core continual-learning metrics, namely, learning performance and memory usage, are analyzed only empirically. Theoretical discussion does not connect quantization distortion to these global measures. If such an analysis is intractable, the paper should explain why; otherwise, incorporating at least approximate analytical links between quantization distortion and forgetting performance would significantly strengthen the contribution."
>
> ---
>
> We agree that forgetting and memory usage are the core continual learning metrics, and appreciate the opportunity to clarify the purpose and scope of our theoretical analysis. Our intent is not to claim that quantization distortion directly predicts forgetting. Rather, the goal of the analysis is to characterize how quantization perturbs the projected update direction in GPM, i.e., how much error QGPM introduces at each SGD step due to quantized projection. This provides a mechanistic understanding of how quantization may affect the optimization path.
>
> In particular, Theorem 3.2, analyzes how the full-precision and quantized GPMs project the same incoming gradient $g$. Let $M_o$ and $M_e$ denote the orthonormal bases of the full-precision and quantized GPMs; the corresponding projection operators are $P_o = M_o M_o^\top$ and $P_e = M_e M_e^\top$. The projected gradients are $\hat g_o = (I - P_o)g$ and $\hat g_e = (I - P_e)g$. Their difference is reflective of the distortion introduced by quantized projection, $\delta g := \hat g_e - \hat g_o = (P_o - P_e)g$. We show that the expected $l_2$ norm of this error is given by
>
> $\mathbb{E}\big[\|\hat g_o - \hat g_e\|_2\big] = m\sigma^2 \|g\|_2$,
>
> where $m$ is the number of stored bases and $\sigma^2$ is the per-basis quantization noise level.
>
> Given a learning rate $\eta$, the full-precision GPM update and the QGPM update from the same parameter vector $w$ are
>
> $w^{\mathrm{FP}} = w - \eta \hat g_o$
>
> $ w^{\mathrm{Q}} = w - \eta \hat g_e = w^{\mathrm{FP}} - \eta \delta g.$
>
> Thus, the per-step deviation between the two trajectories is
>
> $\Delta := w^{\mathrm{Q}} - w^{\mathrm{FP}} = -\eta\delta g,
> \qquad
> \mathbb{E}\big[\|\Delta\|_2\big]
> = \eta \mathbb{E}\big[\|\delta g\|_2\big]
> = \eta m\sigma^2 \|g\|_2.$
>
> Over multiple SGD steps $k$ involved in entire training phase of new task, the accumulated deviation is
>
> $\sum_k \mathbb{E}\big[\|\Delta_k\|_2\big]
> = \sum_k \eta_k  m\sigma^2 \|g_k\|_2,$
>
> which shows how the quantization noise level $\sigma^2$ affects the divergence of the QGPM optimization trajectory from the full-precision baseline. It quantitatively connects quantization to optimization path deviation. We will clarify this optimization trajectory-level perspective in the revised manuscript.

---

> ### Author Response · Authors · 2025-11-25
>
> **[Weakness 4]** "Restricted and oversimplified neural network assumptions - Section 3.1 appears to base the analysis on linear layers without bias terms, which limits the generality of the conclusions. In realistic neural networks, nonlinear activations such as ReLU clip all negative activations to zero, fundamentally changing the distribution and thus the quantization distortion behavior. The authors should discuss how such nonlinearities might alter or invalidate the derived results and whether the approach extends beyond this simplified setting."
>
> ---
>
> Our primary goal in Section 3 was to isolate the core mechanism of QGPM by which quantization affects the GPM projection operator. To ensure clarity, we employ a simplified setting in Section 3.1: a linear layer without biases or nonlinearities. This abstraction highlights the essential principles of the original GPM in its most basic form; this is consistent with prior theoretical analyses in continual learning [3]. That said, our analysis naturally extends to generalized settings with nonlinear activations and bias term. To this end, let us denote pre-activation and activation as $Z^l_\tau = W^l_\tau R^l_\tau + b^l_\tau 1^\top,  A^l_\tau = \sigma^l(Z^l_\tau)$, where $\sigma^l(\cdot)$ is the non-linearity of layer $l$. Preserving activations on past inputs implies
>
> $\sigma^l\big(\mathbf{w}^l_{\tau+1} \mathbf{R}^l_\tau + \mathbf{b}^l_{\tau+1} 1^\top\big)
> = \sigma^l\big(\mathbf{w}^l_\tau \mathbf{R}^l_\tau + \mathbf{b}^l_\tau 1^\top\big).$
>
> A sufficient condition for this is
>
> $\mathbf{w}^l_{\tau+1} \mathbf{R}^l_\tau + \mathbf{b}^l_{\tau+1} 1^\top
> = (\mathbf{w}^l_\tau + \Delta \mathbf{w}^l) \mathbf{R}^l_\tau + (\mathbf{b}^l_\tau + \Delta \mathbf{b}^l) 1^\top
> = \mathbf{w}^l_\tau \mathbf{R}^l_\tau + \mathbf{b}^l_\tau 1^\top$
>
> which simplifies to the orthogonality constraint
>
> $\Delta \mathbf{w}^l \mathbf{R}^l_\tau + \Delta \mathbf{b}^l 1^\top = 0.$
>
> Empirical results validate that relaxing the constraint on $\mathbf{b}$ does not hurt continual learning ability because most of the projection energy is carried through the weights $\mathbf{w}^l$. Thus, we arrive at the same conclusion:
> $\Delta \mathbf{w}^l \mathbf{R}^l_\tau = 0.$
> We will update Section 3.1 to explicitly state these assumptions and clarify that our method is compatible with bias terms and nonlinearities.
>
> [3] Gobinda Saha, Isha Garg, and Kaushik Roy. Gradient projection memory for continual learning. In 9th International Conference on Learning Representations, ICLR 2021, Virtual Event, Austria, May 3-7, 2021.

---

### Official Review · Reviewer_FQfP · 2025-10-28

**Soundness:** 3
**Presentation:** 2
**Contribution:** 3
**Rating:** 6
**Confidence:** 4

**Summary:**

The paper proposes new techniques to quantize the core bases stored for gradient projection method (GPM).The paper first show that the deviation between ideal and quantized orthogonal updates grows quadratically with the quantization error. The paper addresses this using 2 techniques: a) an enhanced quantization approach that improves over the NFk via centered inlier normal float Q aimed at reducing influence of outliers during Q, b) quantization error aware gradient projection to adaptively relax orthogonality constraints based on estimated deviation from desired update direction.

**Strengths:**

- The overall approach to enhance GPM with the new quantization approach as well as the use of sparse sketching for updates have sound reasoning and theoretical bounds as shown in the paper.
- Basis wise quantization makes sense and should have a higher precision and can help retain near orthogonality after quantization
- Experimental results and ablation studies have good coverage in terms of datasets, models and baselines.

**Weaknesses:**

The enhanced quantization over NFk makes sense that it is more robust to outlier values. However, there are a few concerns:
1. Distributional assumptions: codebook is derived from a standard normal. If actual data distribution as pointed out in the paper deviates (especially has skewness), the quantization can be suboptimal
2. This is amplified because of the codebook being fixed and not being adaptive to input distribution
3. Compressing tail could also increase error disproportionately for the NF4; in contrast, the CINF4 codebook construction may over emphasize tails

**Questions:**

What are the assumptions made on the data distribution? How does it impact the quantization performance when the distribution changes? (especially as the paper argues that the continual data has a heavy tail).

Why is the codebook kept static? What are the drawbacks?

---

> ### Author Response · Authors · 2025-11-25
>
> **[Weakness 1 and Question 1]** “What are the assumptions made on the data distribution (standard normal)?”
>
> We thank the reviewer for raising the important question about distributional assumptions. Our key modeling assumption is that the quantization targets, i.e., the GPM basis vectors, are approximately Gaussian, **though not necessarily standard normal**. Empirically, we observe that these basis vectors indeed closely follow the Gaussian assumption, as shown in Figure C(a) of the supplementary link (https://anonymous.4open.science/r/Figure-AD42/Fig%20C.png).
>
> Our proposed Centered Inlier Normal Float (CINF) is conceptually based on quantile quantization [1]. Quantile quantization is known to be information-theoretically optimal in the scalar setting because it assigns equal probability mass to each code bin, minimizing average quantization error. However, direct implementation of true quantile quantization in our setting is impractical for two major practical reasons:
>
> - **Quantile estimation cost**: Accurate estimation of empirical quantiles for each tensor (or even each vector) is computationally intensive, especially in the continual learning settings where subspaces (i.e. bases) are constructed for every task on the fly.
>
> - **Storage overhead of per-tensor codebooks**: Fully adaptive quantile quantization would require storing a separate codebook for each tensor/basis, which is incompatible with our goal of memory-efficient GPM subspace compression.
>
> To avoid these pitfalls, we derive a fixed codebook from the standard normal distribution and match the data to the fixed codebook rather than recompute the codebook for every datum. Specifically:
>
> - Our empirical observations confirm that the centered GPM bases exhibit approximately Gaussian distribution.
>
> - By matching the target basis distribution to the distribution of the codebook (via centering and scaling, as described in lines 161–167 of the main paper), we effectively approximate the behaviour of quantile quantization while avoiding the cost of explicit quantile computation and per-basis codebook storage. This design achieves a practical balance by providing the benefits of distribution-aware quantization while avoiding the computational and memory burdens detrimental to continual learning.
>
> [1] T. Dettmers, M. Lewis, S. Shleifer, and L. Zettlemoyer. 8-bit optimizers via block-wise
> quantization. 9th International Conference on Learning Representations, ICLR, 2022.
>
> **[Weakness 3 and Question 2]** “How does it impact the quantization performance when the distribution changes? (especially as the paper argues that the continual data has a heavy tail).”
>
> As noted above, CINF relies on the assumption that the quantization target (i.e., the GPM basis vectors) are Gaussian-like, not that they follow a strict $\mathcal{N}(0, 1)$ distribution. After centering and scaling, CINF continues to perform well as long as the basis entries are approximately Gaussian, even when the underlying data distribution is heavy-tailed. In fact, by focusing quantization resolution on inlier regions and separately handling outliers (outside the codebook), CINF is naturally robust in the heavy-tailed scenarios.
>
>  However, when the target distribution significantly deviates from Gaussian (e.g., when it becomes near-uniform or highly skewed), the performance of CINF can degrade. In such cases, a codebook derived under Gaussian assumption simply does not align well with the actual data, and even naive affine quantization can perform better. Empirically, we find that the bases added in early layers or in the very early phase of training tend to exhibit non-Gaussian behavior (Figure C(b)), because model parameters and representations are not yet well structured. To address this, we adopt a mixed-scheme quantization strategy (detailed in Appendix Section D). Specifically, leveraging Shapiro-Wilk Normality test, we apply affine quantization to bases that deviate from Gaussianity, and apply CINF to quantize bases with approximately Gaussian distribution – typically those added in deeper layers or later in training.

---

> ### Author Response · Authors · 2025-11-25
>
> **[Weakness 2 and Question 3]** "Why is the codebook kept static? What are the drawbacks?"
>
>
> We appreciate the reviewer’s question regarding the choice of using a static codebook in CINF. This decision is directly motivated by our goals of minimizing memory and computational overhead in QGPM.
>
> In our framework, quantized GPM bases must be dequantized at inference time to perform gradient projection operation (line 3, Alg 1). This implies that, for each basis vector, we also need to store enough information to invert the quantization mapping (i.e. dequantization). If we were to use a dynamic / adaptive quantization scheme, each basis vector would require its own codebook so that it can be accurately dequantized. In practice, these per-basis codebooks would often need to be stored in full precision, which would cause the following drawbacks:
>
> - **Higher memory overhead**: The main goal of QGPM is to provide memory efficiency. Storing separate adaptive codebooks may require a non-negligible amount of auxiliary memory, potentially outweighing the benefits of quantization.
>
> - **Higher computational overhead**: Adaptive quantization and dequantization typically requires estimating quantiles and/or fitting codebooks per vector or tensor, which can add significant runtime cost during both training and inference.
>
> By contrast, using a static codebook derived from the standard normal distribution, CINF enables a much more efficient implementation. For each basis vector, we only store three full-precision scalars (the scaling factor $s$, the mean $\mu$, and the orthogonality weight $\lambda$) needed to invert the quantization. Ultimately, using a static codebook helps achieve a desirable trade-off between accuracy of quantization and memory efficiency.

---

### Official Review · Reviewer_cPdv · 2025-10-29

**Soundness:** 3
**Presentation:** 3
**Contribution:** 2
**Rating:** 6
**Confidence:** 4

**Summary:**

This paper proposes Quantized Gradient Projection Memory, a projection-based continual learning framework that compresses stored gradient subspaces via a distribution-aware quantizer, stabilizes learning with a quantization error–aware projection rule, and reduces construction overhead using on-the-fly sparse sketching, achieving near–full precision accuracy with 4–8× memory savings on standard vision benchmarks under matched memory budgets. The work combines clear algorithmic design, theoretical analyses of quantization-induced drift and sketching guarantees, and thorough experiments that show 8-bit QGPM matches GPM while 4-bit QGPM remains competitive with substantial memory reductions, positioning QGPM as a strong, memory-efficient alternative to rehearsal under privacy constraints.

**Strengths:**

1.This work proposes a targeted quantization scheme CINF for GPM bases and a QEA projection rule that adapts orthogonality weights via cosine-error, addressing quantization-induced subspace distortion in a principled way.

2.This work provides theoretical insights into why NF scaling fails under heavy tails, shows quantization-induced drift accumulation, and gives sketching guarantees, with ablations that align with the theory and quantify bitwidth/outlier/QEA effects on ACC and BWT.

3.The results are highly valuable. The ability to achieve nearly full-precision performance with 4x-6x less memory makes GPM a much more viable competitor to rehearsal-based methods, especially in privacy-sensitive domains.

**Weaknesses:**

1.The core GPM framework still accumulates bases with each new task. While QGPM compresses these bases, the total memory still grows monotonically. The experiments are limited to 5, 10, or 20 tasks. It is unclear how QGPM's memory footprint scales in a true lifelong learning scenario (e.g., 50-100+ tasks) compared to a rehearsal method with a fixed memory buffer. A discussion of this scaling trade-off is missing.

2.The method introduces new hyperparameters that appear critical to performance, particularly the QEA scaling factor $\alpha$ and the CINF outlier proportion $p$. Table 4 shows that performance is highly dependent on $\alpha$. Table 9 indicates that $\alpha$ and $p$ are tuned for each dataset and bitwidth. This raises concerns about the practical tuning cost. A more in-depth sensitivity analysis or a more principled method for setting $\alpha$ (e.g., adapting it based on observed error statistics) would improve robustness.

3.CINF stores outliers in full precision. The paper mentions this contributes minor overhead, but this cost is never explicitly quantified. If the distributions are truly heavy-tailed, storing 1-3% (from Table 9) of vectors in FP32 could be a non-negligible memory cost that offsets the 4-bit/8-bit quantization gains. A brief analysis of this would be beneficial.

**Questions:**

1.Could you comment on the memory scaling of QGPM in a long-sequence setting (e.g., 50+ tasks)? While QGPM compresses bases, the number of bases still grows. Is there a crossover point where a fixed-size rehearsal buffer (like ER) becomes more memory-efficient than the growing QGPM?

2.The QEA factor $\alpha$ seems critical but requires tuning per setup. How sensitive is the method to this hyperparameter? Is there a more principled way to set $\alpha$ automatically, perhaps based on the observed mean/max quantization error, to reduce the tuning burden?

3.What percentage of the final QGPM memory footprint (e.g., in the 8QGPM column of Table 1) is consumed by the full-precision outliers stored by CINF? Is this cost consistently negligible across different models and bitwidths?

4.The sparse sketching component appears to be a strict win, reducing SVD time and intermediate memory with no performance loss (Table 12). Is this always the case? Does the $(1 \pm \epsilon)$-subspace embedding (Theorem 3.3)  ever introduce approximation errors that interact negatively with the quantization errors, or is it robust in all tested scenarios?

---

> ### Author Response · Authors · 2025-11-25
>
> **[Question 1 and Weakness 1]** Long-Term Memory Scaling: “Could you comment on the memory scaling of QGPM in a long-sequence setting (e.g., 50+ tasks)? While QGPM compresses bases, the number of bases still grows. Is there a crossover point where a fixed-size rehearsal buffer (like ER) becomes more memory-efficient than the growing QGPM?”
>
> We thank the reviewer for raising this important question, which parallels the memory scalability concern expressed in Weakness 1. While QGPM does accumulate quantized bases, the memory growth is **not linear** in the number of tasks.This is because the gradient subspaces of new tasks are already well approximated by those formed from past tasks, which leads to diminishing numbers of added bases  over time.
>
> To empirically validate this, we conducted a new **50-task experiment** on miniImageNet using ViT-S. As the results below show, QGPM continues to match full-precision performance, showing robust memory-efficient over rehearsal-based methods such as ER given the same memory budget:
>
> | Method | ACC   | BWT   |
> |--------|-------|-------|
> | 8QGPM (50 Tasks)  | 88.21 | -1.14 |
> | ER (50 Tasks)    | 84.23 | -6.16 |
>
> Moreover, the saturation in the number of basis vectors, driven by diminishing subspace novelty, is illustrated in Figure A of the supplementary link (https://anonymous.4open.science/r/Figure-AD42/Fig%20A.png), which tracks basis count of GPM and total GPM memory use over the 50 tasks.
>
> Finally, as already shown in Figure 3(c) of the main paper, QGPM provides a tunable trade-off between memory and performance via the subspace threshold $\varepsilon_{\text{th}}$. Smaller \varepsilon reduces memory usage by limiting basis addition while still maintaining strong performance; this, in turn, makes QGPM scalable and adaptable even in extremely long-sequence scenarios.
>
> **[Question 2 and Weakness 2]** “The method introduces new hyperparameters that appear critical to performance, particularly the QEA scaling factor α and the CINF outlier proportion p. Table 4 shows that performance is highly dependent on α. Table 9 indicates that α and p are tuned for each dataset and bitwidth. This raises concerns about the practical tuning cost. A more in-depth sensitivity analysis or a more principled method for setting  (e.g., adapting it based on observed error statistics) would improve robustness”
>
> We thank the reviewer for raising this point, which echoes the concern brought up in Weakness 2. In practice, the QEA scaling factor $\alpha$ is **not highly sensitive** and can be selected via a **modest 1D search** over a narrow range (typically between 0 and 30). Empirically, we find that:
> - $\alpha = 10$ works well for 8-bit quantization; and
> - $\alpha = 20$ performs reliably for 4-bit quantization.
>
> This choice of $\alpha$ leads to strong performance across a wide range of architectures (including ViT-S) and datasets. While different values of $\alpha$ were used for miniImageNet experiments on ViT-S in Table 9 of main paper, this is not due to specialized tuning but are simply the values chosen for this run without any optimization. Following up on this, we re-ran the miniImageNet experiment using default hyper-parameter values (i.e., $\alpha = 10$ for 8-bit and $\alpha = 20$ for 4-bit). As shown below, the results remain strong and, in fact, even slightly improved:
>
>  | Method | 10 miniimagenet ACC | 10 miniimagenet BWT | 20 miniimagenet ACC | 20 miniimagenet BWT |
> |--------|----------------------|----------------------|----------------------|----------------------|
> | 8QGPM  | 75.72                | -2.38               | 80.97                | -2.94               |
> |        | 2.55MB               |                      | 2.74MB               |                      |
> | 4QGPM  | 75.02                | -3.98               | 80.33                | -3.69               |
> |        | 1.58MB               |                      | 1.72MB               |                      |
>
> To further reduce tuning effort, we propose a simple rule-of-thumb:
>
> (1) Set outlier ratio $p$ based on quantization bitwidth and memory budget. We recommend 1% for 8-bit and 3% for 4-bit quantization, though lower values can be used in more constrained settings.
>
> (2) Perform a 1-D search for $\alpha$ in $[0, 30]$, monitoring
>  final performance (ACC and BWT). As shown in Table 4, the trend is unimodal – performance increases with $\alpha$ up to a point, then gradually decreases. This allows us to easily identify the sweet spot. This enables efficient tuning and suggests the possibility of setting $\alpha$ based on observed quantization error, which we will explore in future work.

---

> ### Author Response · Authors · 2025-11-25
>
> **[Question 3 and Weakness 3]** Memory overhead due to the full-precision outlier: “What percentage of the final QGPM memory footprint (e.g., in the 8QGPM column of Table 1) is consumed by the full-precision outliers stored by CINF? Is this cost consistently negligible across different models and bitwidths?”
>
> We thank the reviewer for this thoughtful question, connected to Weakness 3 that expressed the concern regarding the cost of storing full-precision outliers under heavy-tail distributions. QGPM’s total memory footprint, including quantized inliers, full-precision outliers (stored by CINF), and auxiliary metadata, can be computed using the following expression:
>
> $ \frac{3 \cdot k \cdot 32 + n \cdot k \cdot \frac{100-p}{100}\cdot q + n \cdot k \cdot \frac{p}{100}\cdot 32}{8 \cdot 1024 \cdot 1024} [\operatorname{MB}] $
>
> where:
> - $n$: basis dimension,
> - $k$: number of basis vectors,
> - $p$: outlier proportion (e.g., 1% for 8-bit, 3% for 4-bit),
> - $q$: quantization bitwidth (4 or 8),
>
> The term 3 $\cdot k \cdot 32$ accounts for auxiliary metadata per basis (scale $s$, mean $\mu$, and orthogonality weight $\lambda$), Note that the middle term accounts for quantized inliers, while the last term accounts for full-precision outliers that are stored explicitly.
>
> To isolate the cost of CINF-stored outliers, we derive the **theoretical compression gain** of QGPM over full-precision GPM as:
>
> $ \operatorname{gain} = \frac{n \cdot k \cdot 32}{3\cdot k \cdot 32+n \cdot k \cdot \frac{100-p}{100}\cdot q + n \cdot k \cdot \frac{p}{100}\cdot 32}$
>
> Typically, $k<<n \cdot k$, we approximate the equation as:
>
> $\operatorname{gain} \approx \frac{32}{\frac{100-p}{100}\cdot q + \frac{p}{100}\cdot 32}$
>
> This expression cleanly characterizes how the outlier ratio $p$ and bitwidth $q$ affect compression. As expected:
> - When $p$ = 0, the gain is exactly 4× for 8-bit and 8× for 4-bit quantization.
> - As $p \rightarrow 100$, the gain converges to 1× (i.e., full-precision).
> - For typical QGPM settings ($p = 1\%, q = 8$), the gain is **3.88×**, and for 4-bit with $p$ = 3\%, the gain is **6.61×**.
>
> We visualize this trade-off in **Figure B** of the supplementary link (https://anonymous.4open.science/r/Figure-AD42/Fig%20B.png), where the gain is plotted as a function of outlier proportion $p$ (when $q$ is set to 4 and 8, respectively). The curve clearly shows that, for small $p$, the overhead from storing full-precision outliers remains **consistently negligible** across models and quantization settings, validating the memory efficiency of CINF even under heavy-tailed settings.
>
>
> **[Question 4]** “The sparse sketching component appears to be a strict win, reducing SVD time and intermediate memory with no performance loss (Table 12). Is this always the case? Does the subspace embedding (Theorem 3.3) ever introduce approximation errors that interact negatively with the quantization errors, or is it robust in all tested scenarios?”
>
> We thank the reviewer for this question. Yes, in our experiments sparse sketching consistently provided substantial runtime and memory savings with negligible impact on performance, confirming it as a strict win in terms of wall-clock time and peak RAM.
>
> As shown in Table 12, applying sparse sketching to ViT-S on miniImageNet reduces GPM construction time (115.06s → 10.60s) and the total training time (1768s → 820s), while the performance as quantified by ACC/BWT remains essentially unchanged (72.21/–0.54 without sketching vs. 72.87/–0.34 with sketching).
>
> From a design perspective, we deliberately choose the sketching dimension $r$ to match the representation dimension $n$ (i.e., $r = n$ in Theorem 3.3). This conservative strategy ensures **high fidelity of sketching** – in particular, a $(1 \pm \epsilon)-l_2$ subspace embedding of the original representation matrix. Consequently, the leading singular directions, and hence the GPM bases, are preserved up to a very small distortion. In the reported experiments, we apply sparse sketching together with quantization (QGPM) across all models and datasets (AlexNet, ResNet-18, ViT-S, ViT-B/16, and the NLP encoder) and consistently observe that any approximation error from sketching is dominated by, and does not amplify, quantization effects. Empirically, we have not observed any negative interaction between sketching and quantization: accuracy and BWT with sketching closely match (and sometimes slightly exceed, due to reduced SVD noise) the non-sketched counterparts under the same quantization settings. We will clarify this point in the final version and explicitly state that, with $r = n$, sparse sketching is robust in all tested scenarios and does not introduce measurable degradation on top of CINF-based quantization.

---

> > ### Comment · Reviewer_cPdv · 2025-11-25
> >
> > Thank you for your thoughtful and thorough response to my comments. I appreciate the clarifications you provided, and they have addressed my concerns satisfactorily. As a result, I will keep my positive score.

---

### Official Review · Reviewer_MtXi · 2025-11-01

**Soundness:** 3
**Presentation:** 3
**Contribution:** 3
**Rating:** 6
**Confidence:** 2

**Summary:**

This paper proposes Quantized Gradient Projection Memory (QGPM), a memory-efficient extension of Gradient Projection Memory (GPM) for continual learning. Traditional GPM stores orthonormal subspace bases to protect previously learned knowledge by projecting gradients onto a constraint subspace. However, its memory cost scales with model size and the number of tasks.

QGPM compresses the stored subspace bases using vector quantization, then dequantizes them only during projection. The authors analyze the quantization-induced distortion and theoretically show that projection error grows with both the number of stored bases and quantization noise. To mitigate forgetting caused by this distortion, the paper introduces Quantization Error-Aware Projection, which adaptively relaxes the orthogonality constraint based on cosine-similarity loss between original and quantized bases.

Experiments across standard continual learning benchmarks demonstrate that QGPM significantly reduces memory consumption while retaining competitive performance against state-of-the-art CL methods.

**Strengths:**

1. Clear motivation for memory efficiency in continual learning by reducing subspace storage cost.

2. Theoretical analysis linking quantization noise to projection error, improving understanding of limitations.

3. Empirical results show that the proposed method keeps performance competitive while significantly reducing memory.

**Weaknesses:**

1. Computational overhead may increase due to repeated dequantization and projection steps, especially for large models.

2. Insufficient ablation study. It would be helpful to see more analysis on different quantization levels and their impact on performance.

**Questions:**

I don't have any questions.

---

> ### Author Response · Authors · 2025-11-24
> **Clarifications on runtime overhead and quantization-level ablations**
>
> **[Weakness 1]** Computational overhead may increase due to repeated dequantization and projection steps, especially for large models.
>
> We appreciate the reviewer’s concern. However, please note that our empirical runtime measurements already include the full cost of quantization and dequantization. In particular, Figure 2(b) shows the end-to-end training time of 8QGPM that accounts for all aspects of the quantization, dequantization and projection steps. Comparing 8QGPM to full precision GPM, GPM‑FP (which performs no quantization), shows that the added computational overhead from the steps introduced by our method is marginal across all tested architectures. Moreover, for larger models such as ViT‑S, the runtime savings from on‑the‑fly sparse sketching exceed the quantization overhead; as a result, 8QGPM with sketching actually trains faster than full‑precision GPM‑FP in practice. We will clarify this point in the revision and explicitly state that Figure 2(b) reflects end-to-end training time, including all quantization-related steps. We appreciate the reviewer’s suggestion to highlight it more clearly.
>
> **[Weakness 2]** Insufficient ablation study. It would be helpful to see more analysis on different quantization levels and their impact on performance.
>
> We thank the reviewer for this suggestion. The impact of quantization bitwidth on QGPM performance is already extensively analyzed in the main paper. In particular, Table 2 on page 8 reports the effect of quantization bitwidth on both the quantization error and the resulting performance of continual learning. Specifically, we evaluate 4-, 5-, 6-, and 8-bit QGPM under fixed hyperparameters (no QEA, no outlier retention) and report: (1) quantization error metrics, including the average ($e_{\text{avg}}$) and maximum ($e_{\text{max}}$) cosine distance between original and quantized basis vectors; and (2) downstream performance metrics, including final accuracy (ACC) and backward transfer (BWT). As expected, increasing bitwidth consistently reduces quantization distortion and improves learning performance. For example, 4-bit QGPM yields high distortion (e_{\text{avg}} = 0.544) and suffers high forgetting (BWT = -31.27), while 8-bit QGPM achieves near-full-precision accuracy (65.01%) with minimal forgetting (BWT = +0.61). We will clarify in the revision that Table 2 provides detailed ablation study over quantization levels, and appreciate the reviewer’s prompting us to make these results more visible.

---

### Author Response · Authors · 2025-12-03

We sincerely thank the Area Chair and all reviewers for their time, thoughtful evaluations, and constructive feedback on our paper. Below, we summarize the key clarifications and revisions that will be incorporated into the final version of our manuscript.

---

**1. Overall contribution and strengths**

We are encouraged that all reviewers recognized the value of our framework, which enables Gradient Projection Memory (GPM) to operate in memory- and privacy-constrained regimes. Our contributions include:

- **CINF**, a distribution-aware scheme for quantizing GPM bases that significantly reduces required storage while minimizing distortion;
- **QEA**, a quantization-aware projection mechanism providing robustness to quantization errors; and
- **Sparse sketching**, an efficient on-the-fly GPM construction that enables scalability to large-scale architectures such as ViTs.

As the reviewers recognized, QGPM achieves **near-full-precision accuracy** while significantly reducing memory across a wide range of datasets and models, which makes it competitive or superior to rehearsal-based methods under matched memory budgets.

---

**2. Clarifications of methodology and theory**

- **Quantization theory & “information-theoretic optimality” (Reviewer 8rDA, FQfP).** We clarified that we do not claim CINF to be globally optimal; our use of “information-theoretically optimal” referred to classical results on **quantile-based scalar quantization**, and how NFk approximates this. We explicitly cite follow-up work showing NF4 is an efficient approximation rather than a true rate-distortion (RD) optimum, and we rephrase our claims accordingly.

- **Distributional assumptions and static codebook (Reviewer FQfP)**. We clarified that our assumption for CINF quantization is that GPM bases are approximately Gaussian, not strictly standard normal. CINF uses a fixed normal-derived codebook as a practical surrogate for quantile quantization, avoiding the computational and memory burden of per-basis quantile estimation and codebook storage. For bases that are clearly non-Gaussian (e.g., early layers / early tasks), we already implemented a mixed quantization scheme (affine vs. CINF) based on normality tests as described in Section D of Appendix.

- **Role of theory vs. CL metrics (Reviewer 8rDA)**. Theorem 3.1 explains why NFk’s max-based normalization fails on high-dimensional SVD bases, motivating CINF’s inlier-quantile scaling. Theorem 3.2 connects quantization noise to **optimization trajectory drift**, bounding the deviation between QGPM and full-precision GPM updates. We clarified that the theoretical goal is to analyze how quantization perturbs projected gradients, rather than directly linking distortion to continual learning phenomena such as forgetting.

- **Why no vector quantization (Reviewer 8rDA)**. We explained that VQ is ill-suited for our setting: the objects being quantized are **orthonormal basis vectors** stored for projection. Sharing codewords across bases would collapse directional information, while online codebook updates would cause codeword drift and extra memory, undermining both projection quality and compression capability. Scalar, basis-wise quantization aligns better with GPM’s geometric structure and preserves its projection properties.

- **Network assumptions (Reviewer 8rDA)**. We clarified that the linear, no-bias model in Section 3.1 is a deliberate simplification to illustrate the core mechanism of QGPM – specifically, how quantization perturbs projected gradients. This is consistent with prior GPM works, which also use simplified settings to isolate and highlight fundamental principles. We demonstrate that the same orthogonality constraints arise even when including biases and nonlinearities, and will include this derivation and discussion in the revised manuscript.

---

---

> ### Author Response · Authors · 2025-12-03
>
> **3. Addressing experimental concerns**
>
> - **Runtime and overhead (Reviewer MtXi)**. We clarified that Figure 2(b) already reports **end‑to‑end training time**, including quantization, dequantization, and projection. The additional overhead of QGPM over GPM‑FP is marginal across all architectures; for large models like ViT‑S, the savings from sparse sketching actually make QGPM faster than GPM‑FP. We will state this explicitly in the paper.
>
> - **Ablations over quantization levels (Reviewer MtXi)**. Table 2 already provides a detailed bitwidth ablation (4/5/6/8 bits) under fixed hyperparameters, reporting both quantization errors ($e_\text{avg}$, $e_\text{max}$) and performance metrics (ACC/BWT). We will more clearly highlight that this table addresses the requested bitwidth sensitivity analysis.
>
> - **Long-sequence scalability (Reviewer cPdv)**. In response to concerns about memory growth on longer task sequence (e.g., 50 tasks), we ran a new **50-task miniImageNet / ViT-S** experiment. QGPM maintains near-full-precision performance and remains more memory-efficient than ER under equal memory budgets. A new figure (Fig. A in the supplemental link) shows that the number of bases and therefore total GPM memory **saturates** over time as subspace novelty diminishes. We also emphasize that the subspace threshold $\varepsilon_{\text{th}}$ provides a tunable memory-accuracy trade-off (Fig. 3(c)), enabling scalable operation even in long task sequences.
>
> - **Hyperparameter robustness (Reviewer cPdv)**. We re-ran the miniImageNet experiments using **unified default** $\alpha$ values ($\alpha$=10 for 8-bit, $\alpha$=20 for 4-bit) and observed equally strong or slightly improved results. This confirms that $\alpha$ is not highly sensitive and does not require per-experiment tuning. We also provide a simple rule-of-thumb to choose $\alpha$: select outlier ratio $p$ based on bitwidth and memory budget (≈1% for 8-bit, ≈3% for 4-bit), then sweep $\alpha$ over a moderate range (e.g., [0, 30]). Table 4’s unimodal ACC/BWT curves confirm that this 1D search is effective and stable.
>
> - **Outlier memory cost (Reviewer cPdv)**. We derived an explicit formula for QGPM memory overhead and a closed-form **compression gain** over full-precision GPM as a function of outlier portion $p$ and quantization bitwidth $q$. For typical QGPM settings (8-bit with $p$=1\%, 4-bit with $p$=3\%), the theoretical gains are 3.88× and 6.61× respectively. Fig. B in the supplemental link shows that, in the regime we use, the memory cost of full-precision outliers is **consistently negligible**.
>
> - **Sparse sketching robustness (Reviewer cPdv)**. We clarified that the sketch dimension $r$ is set to match the representation dimension $n$ in order to obtain a conservative $(1\pm\epsilon)-l_2$ subspace embedding. Across all models and datasets, sketching plus QGPM yields nearly identical or slightly improved ACC/BWT, while significantly reducing SVD cost and total training time (Table 12). We have not observed any negative interaction between sketching and quantization.
>
> ---
>
> We again sincerely thank the anonymous reviewers for their thoughtful feedback and constructive suggestions, which have helped significantly improve the quality of our work. We hope that the  clarifications and new experimental results fully address the raised concerns and further highlighted the contribution of QGPM as a practical, theoretically grounded approach to memory-efficient continual learning.

---

### Meta-Review · Area_Chair_E26x · 2025-12-19

**Summary:**

Initial concerns centered on overstated theoretical claims around “information-theoretic optimality,” unclear distributional assumptions behind the fixed normal-derived codebook, and a perceived disconnect between the quantization analysis and continual-learning metrics, along with questions about vector quantization, simplified network assumptions, and practical scalability/tuning (runtime overhead, long task sequences, hyperparameter sensitivity, and outlier memory cost). The authors addressed these by rephrasing the optimality claim to explicitly scope it to classical scalar quantile quantization and citing follow-up work on NF4 as an approximation, clarifying that CINF assumes approximately Gaussian (not strictly standard normal) GPM bases and uses a static codebook for memory/compute reasons with a mixed scheme for non-Gaussian bases, and reframing the theory as analyzing quantization-induced projection/trajectory drift rather than directly predicting forgetting. They further strengthened the empirical story with explicit end-to-end runtime clarification, bitwidth ablations, a new 50-task miniImageNet/ViT-S experiment showing saturation of basis growth, unified default hyperparameter settings and a stable 1D search rule-of-thumb, and a closed-form accounting of outlier overhead and compression gains, plus clarifications on why vector quantization is ill-suited for orthonormal bases and how the analysis extends beyond linear no-bias models. Reviewers generally found the responses thorough and maintained (or kept) positive scores. The paper’s contribution, making GPM viable in memory- and privacy-constrained regimes via distribution-aware basis quantization, quantization-error-aware projection, and scalable sparse sketching, was consistently viewed as novel and practically valuable, with remaining limitations considered acceptable given the clarified scope and strengthened evidence.

**Reviewer Concerns:**

The reviewers initially raised several key concerns:

(1) theoretical claims were overstated (e.g., “information-theoretically optimal”) and the novelty/role of the quantization theory was unclear;

(2) distributional assumptions (Gaussianity) and the choice of a static normal-derived codebook could be suboptimal under skew/heavy tails;

(3) a disconnect between theoretical distortion analysis and core continual-learning outcomes (ACC/BWT/forgetting), and oversimplified network assumptions (linear, no-bias);

(4) lack of justification for not using vector quantization / stochastic quantization;

(5) practical concerns about scalability and robustness: runtime overhead, long-sequence memory scaling (50+ tasks), hyperparameter sensitivity (e.g., QEA factor and outlier ratio), and whether full-precision outliers materially reduce compression gains.

**Reviewer Scores:**

In response, the authors provided substantive clarifications and new supporting evidence. They explicitly revised the “information-theoretic optimality” wording to avoid any claim of global RD optimality, grounding it instead in classical scalar quantile-quantization results and adding citations that characterize NF4 as an efficient approximation rather than a true optimum. They clarified that CINF assumes GPM bases are approximately Gaussian (not necessarily standard normal), and that the static codebook is a deliberate compute/memory trade-off; for clearly non-Gaussian bases (often early layers/early tasks) they employ a mixed quantization strategy triggered by normality tests. They clarified the purpose of the theory: Theorem 3.1 diagnoses NFk’s max-normalization failure on high-dimensional SVD bases and motivates inlier-quantile scaling, while Theorem 3.2 links quantization noise to per-step projected-gradient error and optimization trajectory drift bounds, without claiming a direct closed-form link to forgetting metrics. They further justified the omission of vector quantization by arguing it conflicts with the geometric requirements of storing orthonormal basis directions (codeword reuse collapses directions, and online codebook drift adds memory and instability), making basis-wise scalar quantization better aligned with GPM’s projection properties. On the empirical side, they clarified that runtime plots already measure end-to-end cost (including quantize/dequantize/projection) and that sparse sketching can make QGPM faster than full-precision GPM for large models. They highlighted existing bitwidth ablations (4/5/6/8 bits) and added a new 50-task miniImageNet/ViT-S experiment demonstrating near-full-precision performance and saturation of basis growth, addressing long-horizon scaling. They also strengthened robustness claims via unified default hyperparameters, a practical rule-of-thumb plus stable 1D search behavior, and an explicit formula quantifying outlier overhead and compression gains, showing outliers remain negligible under typical settings. Reviewers MtXi, cPdv, and FQfP indicated their concerns were addressed and maintained positive scores; the revised positioning and additional experiments substantially improved clarity, rigor, and practical credibility. The work’s strengths, distribution-aware compression of GPM memory, quantization-aware projection to preserve learning dynamics, and scalable sparse sketching enabling large architectures, were repeatedly recognized as valuable contributions for privacy-constrained continual learning.

---

### Decision · Program_Chairs · 2026-01-26

Accept (Poster)